# Dynamical Isometry for Residual Networks

## Abstract

The training success, training speed and generalization ability of neural networks rely crucially on the choice of random parameter initialization. It has been shown for multiple architectures that initial dynamical isometry is particularly advantageous. Known initialization schemes for residual blocks, however, miss this property and suffer from degrading separability of different inputs for increasing depth and instability without Batch Normalization or lack feature diversity. We propose a random initialization scheme, Risotto, that achieves perfect dynamical isometry for residual networks with ReLU activation functions even for finite depth and width. It balances the contributions of the residual and skip branches unlike other schemes, which initially bias towards the skip connections. In experiments, we demonstrate that in most cases our approach outperforms initialization schemes proposed to make Batch Normalization obsolete, including Fixup and SkipInit, and facilitates stable training. Also in combination with Batch Normalization, we find that Risotto often achieves the overall best result.

## 1 Introduction

Random initialization of weights in a neural network play a crucial role in determining the final performance of the network. This effect becomes even more pronounced for very deep models that seem to be able to solve many complex tasks more effectively. An important building block of many models are residual blocks He et al. (2016), in which skip connections between non-consecutive layers are added to ease signal propagation (Balduzzi et al., 2017) and allow for faster training. ResNets, which consist of multiple residual blocks, have since become a popular center piece of many deep learning applications (Bello et al., 2021).

Batch Normalization (BN) (Ioffe & Szegedy, 2015) is a key ingredient to train ResNets on large datasets. It allows training with larger learning rates, often improves generalization, and makes the training success robust to different choices of parameter initializations. It has furthermore been shown to smoothen the loss landscape (Santurkar et al., 2018) and to improve signal propagation (De & Smith, 2020). However, BN has also several drawbacks: It breaks the independence of samples in a minibatch and adds considerable computational costs. Sufficiently large batch sizes to compute robust statistics can be infeasible if the input data requires a lot of memory. Moreover, BN also prevents adversarial training (Wang et al., 2022). For that reason, it is still an active area of research to find alternatives to BN Zhang et al. (2018); Brock et al. (2021b). A combinations of Scaled Weight Standardization and gradient clipping has recently outperformed BN (Brock et al., 2021b). However, a random parameter initialization scheme that can achieve all the benefits of BN is still an open problem. An initialization scheme allows deep learning systems the flexibility to drop in to existing setups without modifying pipelines. For that reason, it is still necessary to develop initialization schemes that enable learning very deep neural network models without normalization or standardization methods.

A direction of research pioneered by Saxe et al. (2013); Pennington et al. (2017) has analyzed the signal propagation through randomly parameterized neural networks in the infinite width limit using random matrix theory. They have argued that parameter initialization approaches that have the *dynamical isometry* (DI) property avoid exploding or vanishing gradients, as the singular values of the input-output Jacobian are close to unity. DI is key to stable and fast training (Du et al., 2019; Hu et al., 2020). While Pennington et al. (2017) showed that it is not possible to achieve DI in networks with ReLU activations with independent weights or orthogonal weight matrices, Burkholz & Dubatovka (2019); Balduzzi et al. (2017) derived a way to attain perfect DI even in finite ReLU

networks by parameter sharing. This approach can also be combined (Blumenfeld et al., 2020; Balduzzi et al., 2017) with orthogonal initialization schemes for convolutional layers (Xiao et al., 2018). The main idea is to design a random initial network that represents a linear isometric map.

We transfer a similar idea to ResNets but have to overcome the additional challenge of integrating residual connections and, in particular, potentially non-trainable identity mappings. In contrast to other ResNet initialization schemes that achieve (approximate) dynamical isometry by initially scaling down the residual connections, we balance the weight of skip and residual connections and thus promote higher initial feature diversity, as highlighted by Fig. 1. We thus propose RISOTTO (**R**esidual dynamical **iso**metry by ini**t**ial **o**rthogonality), an initialization scheme that induces *exact dynamical isometry* (DI) for ResNets (He et al., 2016) with convolutional or fully-connected layers and ReLU activation functions. RISOTTO achieves this for networks of finite

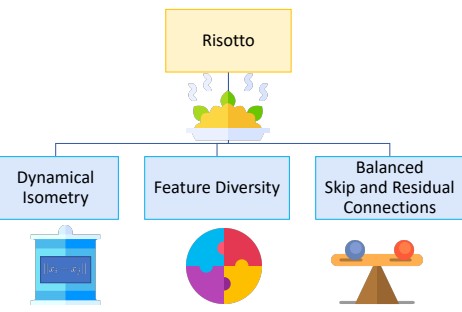

Figure 1: Features of our initialization scheme Risotto. This figure has been designed using images from Flaticon.com.

width and finite depth and not only in expectation but exactly. We provide theoretical and empirical evidence that highlight the advantages of our approach. Remarkably and in contrast to other initialization schemes that aim to improve signal propagation in ResNets, RISOTTO can achieve performance gains even in combination with BN. We hypothesize that this improvement is due to the balanced skip and residual connections. Furthermore, we demonstrate that RISOTTO can successfully train ResNets without BN and achieve the same or better performance than Zhang et al. (2018); Brock et al. (2021b).

## 1.1 CONTRIBUTIONS

- To explain the drawbacks of most initialization schemes for residual blocks, we derive signal propagation results for finite networks without requiring mean field approximations and highlight input separability issues for large depths. Accordingly, activations corresponding to different inputs become more similar for increasing depth which makes it harder for the neural network to distinguish different classes.

- We propose a solution, RISOTTO, which is an initialization scheme for residual blocks that provably achieves dynamical isometry (exactly for finite networks and not only approximately). A residual block is initialized so that it acts as an orthogonal, norm and distance preserving transform.

- In experiments on multiple standard benchmark datasets, we demonstrate that our approach achieves competitive results in comparison with alternatives:

  - We show that RISOTTO facilitates training ResNets without BN or any other normalization method and often outperforms existing BN free methods including Fixup, SkipInit and NF ResNets.
  - It outperforms standard initialization schemes for ResNets with BN on Tiny Imagenet, CIFAR100, and ImageNet.

## 1.2 RELATED WORK

**Preserving Signal Propagation** Random initialization schemes have been designed for a multitude of neural network architectures and activation functions. Early work has focused on the layerwise preservation of average squared signal norms (Glorot & Bengio, 2010; He et al., 2015; Hanin, 2018) and their variance (Hanin & Rolnick, 2018). The mean field theory of infinitely wide networks has also integrated signal covariances into the analysis and further generated practical insights into good choices that avoid exploding or vanishing gradients and enable feature learning (Yang & Hu, 2021) if the parameters are drawn independently (Poole et al., 2016; Raghu et al., 2017; Schoenholz et al., 2017; Yang & Schoenholz, 2017; Xiao et al., 2018). Indirectly, these works demand that the average eigenvalue of the signal input-output Jacobian is steered towards 1. Yet, in this set-up, ReLU activation functions fail to support parameter choices that lead to good trainability of very

deep networks, as outputs corresponding to different inputs become more similar for increasing depth (Poole et al., 2016; Burkholz & Dubatovka, 2019). Yang & Schoenholz (2017) could show that ResNets can mitigate this effect and enable training deeper networks, but also cannot distinguish different inputs eventually.

However, there are exceptions. Balanced networks can improve (Li et al., 2021) interlayer correlations and reduce the variance of the output. A more effective option is to remove the contribution of the residual part entirely as proposed in successful ResNet initialization schemes like Fixup (Zhang et al., 2018), SkipInit (De & Smith, 2020), or ISONet (Qi et al., 2020). This, however, limits significantly the initial feature diversity that is usually crucial for the training success (Blumenfeld et al., 2020) and scales down the residual connections. A way to address this issue for other architectures with ReLUs like fully-connected (Burkholz & Dubatovka, 2019) and convolutional (Balduzzi et al., 2017) layers is a looks-linear weight matrix structure (Shang et al., 2016). This idea has not been transferred to residual blocks yet but has the advantage that it can be combined with orthogonal submatrices. These matrices induce perfect dynamical isometry (Saxe et al., 2013; Mishkin & Matas, 2015; Poole et al., 2016; Pennington et al., 2017), meaning that the eigenvalues of the initial input-output Jacobian are identical to $1$ or $-1$ and not just close to unity on average. This property has been shown to enable the training of very deep neural networks (Xiao et al., 2018) and can improve their generalization ability (Hayase & Karakida, 2021) and training speed Pennington et al. (2017; 2018). ResNets equipped with ReLUs can currently only achieve this property approximately and without a practical initialization scheme (Tarnowski et al., 2019) or with reduced feature diversity (Blumenfeld et al., 2020) and potential training instabilities (Zhang et al., 2018; De & Smith, 2020).

**ResNet Initialization Approaches**  Fixup (Zhang et al., 2018), SkipInit (De & Smith, 2020), ISONet (Qi et al., 2020), and ReZero (Bachlechner et al., 2021) have been designed to enable training without requiring BN, yet, can usually not achieve equal performance. Training data informed approaches have also been successful (Zhu et al., 2021; Dauphin & Schoenholz, 2019) but they require computing the gradient of the input minibatches. Yet, most methods only work well in combination with BN (Ioffe & Szegedy, 2015), as it seems to improve ill conditioned initializations (Glorot & Bengio, 2010; He et al., 2016) according to Bjorck et al. (2018), allows training with larger learning rates (Santurkar et al., 2018), and might initially bias the residual block towards the identity enabling signal to flow through De & Smith (2020). The additional computational and memory costs of BN, however, have motivated research on alternatives including different normalization methods (Wu & He, 2018; Salimans & Kingma, 2016; Ulyanov et al., 2016). Only recently has it been possible to outperform BN in generalization performance using scaled weight standardization and gradient clipping (Brock et al., 2021b;a), but this requires careful hyperparameter tuning. In experiments, we compare our initialization proposal RISOTTO with all three approaches: normalization free methods, BN, and normalization alternatives (e.g NF ResNet).

## 2  RESNET INITIALIZATION

### 2.1  BACKGROUND AND NOTATION

The object of our study is a general residual network that is defined by

$$\boldsymbol{z}^0 := \mathbf{W}^0 * \boldsymbol{x}, \quad \boldsymbol{x}^l = \phi(\boldsymbol{z}^{l-1}), \quad \boldsymbol{z}^l = \alpha_l f^l(\boldsymbol{x}^l) + \beta_l h^l(\boldsymbol{x}^l); \quad \boldsymbol{z}^{\text{out}} := \mathbf{W}^{\text{out}} P(\boldsymbol{x}^L) \quad (1)$$

for $1 \leq l \leq L$. $P(.)$ denotes an optional pooling operation like maxpool or average pool, $f(.)$ residual connections, and $h(.)$ the skip connections, which usually represent an identity mapping or a projection. For simplicity, we assume in our derivations and arguments that these functions are parameterized as $f^l(\boldsymbol{x}^l) = \mathbf{W}_2^l * \phi(\mathbf{W}_1^l * \boldsymbol{x}^l + \boldsymbol{b}_1^l) + \boldsymbol{b}_2^l$ and $h^l(\boldsymbol{x}^l) = \mathbf{W}_{\text{skip}}^l * \boldsymbol{x}^l + \boldsymbol{b}_{\text{skip}}^l$ ($*$ denotes convolution), but our arguments also transfer to residual blocks in which more than one layer is skipped. Optionally, batch normalization (BN) layers are placed before or after the nonlinear activation function $\phi(\cdot)$. We focus on ReLUs $\phi(x) = \max\{0, x\}$ (Krizhevsky et al., 2012), which are among the most commonly used activation functions in practice. All biases $\boldsymbol{b}_2^l \in \mathbb{R}^{N_{l+1}}$, $\boldsymbol{b}_1^l \in \mathbb{R}^{N_{m_l}}$, and $\boldsymbol{b}_{\text{skip}}^l \in \mathbb{R}^{N_l}$ are assumed to be trainable and set initially to zero. We ignore them in the following, since we are primarily interested in the neuron states and signal propagation at initialization. The parameters $\alpha$ and $\beta$ balance the contribution of the skip and the residual branch, respectively. Note that $\alpha$ is a trainable parameter, while $\beta$ is just mentioned for convenience to simplify the comparison with standard He initialization approaches (He et al., 2015). Both parameters could also be

integrated into the weight parameters $\mathbf{W}_2^l \in \mathbb{R}^{N_{l+1} \times N_{m_l} \times k_{2,1}^l \times k_{2,2}^l}$, $\mathbf{W}_1^l \in \mathbb{R}^{N_{m_l} \times N_l \times k_{1,2}^l \times k_{1,2}^l}$, and $\mathbf{W}_{\text{skip}}^l \in \mathbb{R}^{N_{l+1} \times N_l \times 1 \times 1}$, but they make the discussion of different initialization schemes more convenient and simplify the comparison with standard He initialization approaches (He et al., 2015).

**Residual Blocks** Following the definition by He et al. (2015), we distinguish two types of residual blocks, Type B and Type C (see Figure 2a), which differ in the choice of $\mathbf{W}_{\text{skip}}^l$. The Type C residual block is defined as $z^l = \alpha f^l(x^l) + h^l(x^l)$ so that shortcuts $h(.)$ are projections with a $1 \times 1$ kernel with trainable parameters. The type B residual block has identity skip connections $z^l = \alpha f^l(x^l) + x^l$. Thus, $\mathbf{W}_{\text{skip}}^l$ represents the identity and is not trainable.

## 2.2 SIGNAL PROPAGATION FOR NORMAL RESNET INITIALIZATION

Most initialization methods for ResNets draw weight entries independently at random, including Fixup and SkipInit. To simplify the theoretical analysis of the induced random networks and to highlight the shortcomings of the independence assumption, we assume:

**Definition 2.1** (Normally Distributed ResNet Parameters). *All biases are initialized as zero and all weight matrix entries are independently normally distributed with*

$$w_{ij,2}^l \sim \mathcal{N}\left(0, \sigma_{l,2}^2\right), w_{ij,1}^l \sim \mathcal{N}\left(0, \sigma_{l,1}^2\right), \text{ and } w_{ij,skip}^l \sim \mathcal{N}\left(0, \sigma_{l,skip}^2\right).$$

Most studies further focus on special cases of the following set of parameter choices.

**Definition 2.2** (Normal ResNet Initialization). *The choice* $\sigma_{l,1} = \sqrt{\frac{2}{N_{m_l} k_{1,1}^l k_{1,2}^l}}$, $\sigma_{l,2} = \sqrt{\frac{2}{N_{l+1} k_{2,1}^l k_{2,2}^l}}$, $\sigma_{l,skip} = \sqrt{\frac{2}{N_{l+1}}}$ *as used in Definition 2.1 and* $\alpha_l, \beta_l \geq 0$ *that fulfill* $\alpha_l^2 + \beta_l^2 = 1$.

Another common choice is $\mathbf{W}_{\text{skip}} = \mathbb{I}$ instead of random entries. If $\beta_l = 1$, sometimes also $\alpha_l \neq 0$ is still common if it accounts for the depth $L$ of the network. In case $\alpha_l$ and $\beta_l$ are the same for each layer we drop the subscript $l$. For instance, Fixup (Zhang et al., 2018) and SkipInit (De & Smith, 2020) satisfy the above condition with $\alpha = 0$ and $\beta = 1$. De & Smith (2020) argue that BN also suppresses the residual branch effectively. However, in combination with He initialization (He et al., 2015) it becomes more similar to $\alpha = \beta = \sqrt{0.5}$. Li et al. (2021) study the case of free $\alpha_l$ but focus their analysis on identity mappings $\mathbf{W}_1^l = \mathbb{I}$ and $\mathbf{W}_{\text{skip}}^l = \mathbb{I}$.

As other theoretical work, we focus our following investigations on fully-connected layers to simplify the exposition. Similar insights would transfer to convolutional layers but would require extra effort (Yang & Schoenholz, 2017). The motivation for the general choice in Definition 2.2 is that it ensures that the average squared l2-norm of the neuron states is identical in every layer. This has been shown by Li et al. (2021) for the special choice $\mathbf{W}_1^l = \mathbb{I}$ and $\mathbf{W}_{\text{skip}}^l = \mathbb{I}$, $\beta = 1$ and by (Yang & Schoenholz, 2017) in the mean field limit with a missing ReLU so that $x^l = z^{l-1}$. (Hanin & Rolnick, 2018) has also observed for $\mathbf{W}_{\text{skip}}^l = I$ and $\beta = 1$ that the squared signal norm increases in $\sum_l \alpha_l$. For completeness, we present the most general case next and prove it in the appendix.

**Theorem 2.3** (Norm preservation). *Let a neural network consist of fully-connected residual blocks as defined by Equ. (1) that start with a fully-connected layer at the beginning* $\mathbf{W}^0$, *which contains* $N_1$ *output channels. Assume that all biases are initialized as 0 and that all weight matrix entries are independently normally distributed with* $w_{ij,2}^l \sim \mathcal{N}\left(0, \sigma_{l,2}^2\right)$, $w_{ij,1}^l \sim \mathcal{N}\left(0, \sigma_{l,1}^2\right)$, *and* $w_{ij,skip}^l \sim \mathcal{N}\left(0, \sigma_{l,skip}^2\right)$. *Then the expected squared norm of the output after one fully-connected layer and* $L$ *residual blocks applied to input* $x$ *is given by*

$$\mathbb{E}\left(\|x^L\|^2\right) = \frac{N_1}{2} \sigma_0^2 \prod_{l=1}^{L-1} \frac{N_{l+1}}{2} \left(\alpha_l^2 \sigma_{l,2}^2 \sigma_{l,1}^2 \frac{N_{m_l}}{2} + \beta_l^2 \sigma_{l,skip}^2\right) \|x\|^2.$$

Note that this result does not rely on any (mean field) approximations and applies also to other parameter distributions that have zero mean and are symmetric around zero. Inserting the parameters of Definition 2.1 for fully-connected networks with $k = 1$ leads to the following insight that explains why this is the preferred initialization choice.

**Insight 2.4** (Norm preserving initialization). *Acccording to Theorem 2.3, the normal ResNet initialization (Definition 2.2) preserves the average squared signal norm for arbitrary depth* $L$.

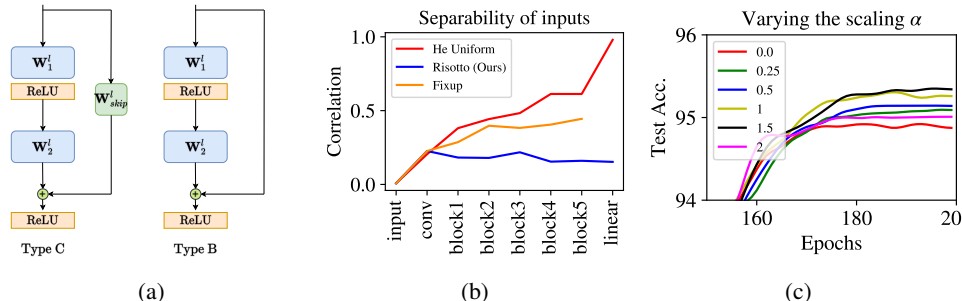

(a)  (b)  (c)

Figure 2: $(a)$ The two types of considered residual blocks. In Type C the skip connection is a projection with a $1 \times 1$ kernel while in Type B the input is directly added to the residual block via the skip connection. Both these blocks have been described by He et al. (2016). $(b)$ The correlation between two inputs for different initializations as they pass through a residual network consisting of a convolution filter followed by 5 residual blocks (Type C), an average pool, and a linear layer on CIFAR10. Only RISOTTO maintains constant correlations after each residual block while it increases for the other initializations with depth. $(c)$ Balancing skip and residual connections: Performance of RISOTTO for different values of alpha ($\alpha$) for ResNet 18 (C) on CIFAR10. $\alpha = 0$ is equivalent to SkipInit and achieves the lowest accuracy. The better performing $\alpha = 1$ is implemented in Risotto.

Even though this initialization setting is able to avoid exploding or vanishing signals, it still induces considerable issues, as the analysis of the joint signal corresponding to different inputs reveals. According to the next theorem, the signal covariance fulfills a layerwise recurrence relationship that leads to the observation that signals become more similar with increasing depth.

**Theorem 2.5** (Layerwise signal covariance). *Let a fully-connected residual block be given as defined by Eq. (1) with random parameters according to Definition 2.2. Let $\boldsymbol{x}^{l+1}$ denote the neuron states of Layer $l + 1$ for input $x$ and $\tilde{\boldsymbol{x}}^{l+1}$ the same neurons but for input $\tilde{x}$. Then their covariance given all parameters of the previous layers is given as $\mathbb{E}_l \left( \langle \boldsymbol{x}^{l+1}, \tilde{\boldsymbol{x}}^{l+1} \rangle \right)$*

$$\geq \frac{1}{4} \frac{N_{l+1}}{2} \left( \alpha^2 \sigma_{l,2}^2 \sigma_{l,1}^2 \frac{N_{m_l}}{2} + 2\beta^2 \sigma_{l,skip}^2 \right) \langle \boldsymbol{x}^l, \tilde{\boldsymbol{x}}^l \rangle + \frac{c}{4} \alpha^2 N_{l+1} \sigma_{l,2}^2 \sigma_{l,1}^2 N_{m_l} \left\| \boldsymbol{x}^l \right\| \left\| \tilde{\boldsymbol{x}}^l \right\| \tag{2}$$

$$+ \mathbb{E}_{\mathbf{W}_1^l} \left( \sqrt{\left( \alpha^2 \sigma_{l,2}^2 \left\| \phi(\mathbf{W}_1^l \boldsymbol{x}^l) \right\|^2 + \beta^2 \sigma_{l,skip}^2 \left\| \boldsymbol{x}^l \right\|^2 \right) \left( \alpha^2 \sigma_{l,2}^2 \left\| \phi(\mathbf{W}_1^l \tilde{\boldsymbol{x}}^l) \right\|^2 + \beta^2 \sigma_{l,skip}^2 \left\| \tilde{\boldsymbol{x}}^l \right\|^2 \right)} \right),$$

*where the expectation $\mathbb{E}_l$ is taken with respect to the initial parameters $\mathbf{W}_2^l$, $\mathbf{W}_1^l$, and $\mathbf{W}_{skip}^l$ and the constant $c$ fulfills $0.24 \leq c \leq 0.25$.*

Note that this statement holds even for finite networks. To clarify what that means for the separability of inputs, we have to compute the expectation with respect to the parameters of $\mathbf{W}_1$. To gain an intuition, we employ an approximation that holds for a wide intermediary network.

**Insight 2.6** (Covariance of signal for different inputs increases with depth). *Let a fully-connected ResNet with random parameters as in Definition 2.2 be given. It follows from Theorem 2.5 that the outputs corresponding to different inputs become more difficult to distinguish for increasing depth $L$. For simplicity, let us assume that $\|\boldsymbol{x}\| = \|\tilde{\boldsymbol{x}}\| = 1$. Then, in the mean field limit $N_{m_l} \to \infty$, the covariance of the signals is lower bounded by*

$$\mathbb{E} \left( \langle \boldsymbol{x}^L, \tilde{\boldsymbol{x}}^L \rangle \right) \geq \gamma_1^L \langle \boldsymbol{x}, \tilde{\boldsymbol{x}} \rangle + \gamma_2 \sum_{k=0}^{L-1} \gamma_1^k = \gamma_1^L \langle \boldsymbol{x}, \tilde{\boldsymbol{x}} \rangle + \frac{\gamma_2}{1 - \gamma_1} \left( 1 - \gamma_1^L \right) \tag{3}$$

*for $\gamma_1 = \frac{1+\beta^2}{4} \leq \frac{1}{2}$ and $\gamma_2 = c(\alpha^2 + 2) \approx \frac{\alpha^2}{4} + \frac{1}{2}$ using $E_{l-1} \|\boldsymbol{x}^l\| \|\tilde{\boldsymbol{x}}^l\| \approx 1$.*

Since $\gamma_1 < 1$, the contribution of the original input correlations $\langle \boldsymbol{x}, \tilde{\boldsymbol{x}} \rangle$ vanishes for increasing depth $L$. Meanwhile, by adding constant contribution in every layer, irrespective of the input correlations, $\mathbb{E} \left( \langle \boldsymbol{x}^L, \tilde{\boldsymbol{x}}^L \rangle \right)$ increases with $L$ and converges to the maximum value 1 (or a slightly smaller value in case of smaller width $N_{m_l}$). Thus, deep models essentially map every input to almost the same

output vector, which makes it impossible for the initial network to distinguish different inputs and provide information for meaningful gradients. Fig. 2b demonstrates this trend and compares it with our initialization proposal RISOTTO, which does not suffer from this problem.

While the general trend holds for residual as well as standard fully-connected feed forward networks ($\beta = 0$), interestingly, we still note a mitigation for a strong residual branch ($\beta = 1$). The contribution by the input correlations decreases more slowly and the constant contribution is reduced for larger $\beta$. Thus, residual networks make the training of deeper models feasible, as they were designed to do (He et al., 2016). This observation is in line with the findings of Yang & Schoenholz (2017), which were obtained by mean field approximations for a different case without ReLU after the residual block (so that $\boldsymbol{x}^l = \boldsymbol{z}^{l-1}$). It also explains how ResNet initialization approaches like Fixup (Zhang et al., 2018) and SkipInit (De & Smith, 2020) can be successful in training deep ResNets. They set $\alpha = 0$ and $\beta = 1$. If $\mathbf{W}_{\text{skip}} = \mathbb{I}$, this approach even leads to dynamical isometry but trades it for very limited feature diversity (Blumenfeld et al., 2020) and initially broken residual branch. Figure 2c highlights potential advantages that can be achieved by $\alpha \neq 0$ if the initialization can still maintain dynamical isometry as our proposal RISOTTO.

### 2.3 RISOTTO: ORTHOGONAL INITIALIZATION OF RESNETS FOR DYNAMICAL ISOMETRY

Our main objective is to avoid the highlighted drawbacks of the ResNet initialization schemes that we have discussed in the last section. We aim to not only maintain input correlations on average but exactly and ensure that the input-output Jacobian of our randomly initialized ResNet is an isometry. All its eigenvalues equal thus $1$ or $-1$. In comparison with Fixup and SkipInit, we also seek to increase the feature diversity and allow for arbitrary scaling of the residual versus the skip branch.

**Looks-linear matrix structure** The first step in designing an orthogonal initialization for a residual block is to allow signal to propagate through a ReLU activation without loosing half of the information. This can be achieved with the help of a looks-linear initialization (Shang et al., 2016; Burkholz & Dubatovka, 2019; Balduzzi et al., 2017), which leverages the identity $\boldsymbol{x} = \phi(\boldsymbol{x}) - \phi(-\boldsymbol{x})$. Accordingly, the first layer maps the transformed input to a positive and a negative part. A fully-connected layer is defined by $\boldsymbol{x}^1 = \left[\hat{\boldsymbol{x}}_+^1; \hat{\boldsymbol{x}}_-^1\right] = \phi\left([\mathrm{U}^0; -\mathrm{U}^0]\boldsymbol{x}\right)$ with respect to a submatrix $\mathrm{U}^0$. Note that the difference of both components defines a linear transformation of the input $\hat{\boldsymbol{x}}_+^1 - \hat{\boldsymbol{x}}_-^1 = \mathrm{U}^0\boldsymbol{x}$. Thus, all information about $\mathrm{U}^0\boldsymbol{x}$ is contained in $\boldsymbol{x}^1$. The next layers continue to separate the positive and negative part of a signal. Assuming this structure as input, the next layers $\boldsymbol{x}^{l+1} = \phi(\mathbf{W}^l\boldsymbol{x}^l)$ proceed with the block structure $\mathbf{W}^l = \left[\mathrm{U}^l \quad -\mathrm{U}^l; \mathrm{U}^l \quad -\mathrm{U}^l\right]$. As a consequence, the activations of every layer can be separated into a positive and a negative part as $\boldsymbol{x}^l = \left[\hat{\boldsymbol{x}}_+^l; \hat{\boldsymbol{x}}_-^l\right]$ so that $\left\|\boldsymbol{x}^l\right\| = \left\|\boldsymbol{z}^{l-1}\right\|$. The submatrices $\mathrm{U}^l$ can be specified as in case of a linear neural network. Thus, if they are orthogonal, they induce a neural network with the dynamical isometry property (Burkholz & Dubatovka, 2019). With the help of the Delta Orthogonal initialization (Xiao et al., 2018), the same idea can also be transferred to convolutional layers. Given a matrix $\mathrm{H} \in \mathbb{R}^{N_{l+1} \times N_l}$, a convolutional tensor is defined as $\mathbf{W} \in \mathbb{R}^{N_{l+1} \times N_l \times k_1 \times k_2}$ as $w_{ijk_1'k_2'} = h_{ij}$ if $k_1' = \lfloor k_1/2 \rfloor$ and $k_2' = \lfloor k_2/2 \rfloor$ and $w_{ijk_1'k_2'} = 0$ otherwise. We make frequent use of the combination of the idea behind the Delta Orthogonal initialization and the looks-linear structure.

**Definition 2.7** (Looks-linear structure). *A tensor $\mathbf{W} \in \mathbb{R}^{N_{l+1} \times N_l \times k_1 \times k_2}$ is said to have looks-linear structure with respect to a submatrix $\mathrm{U} \in \mathbb{R}^{\lfloor N_{l+1}/2 \rfloor \times \lfloor N_l/2 \rfloor}$ if*

$$w_{ijk_1'k_2'} = \left\{ \begin{array}{ll} h_{ij} & \text{if } k_1' = \lfloor k_1/2 \rfloor \text{ and } k_2' = \lfloor k_2/2 \rfloor, \\ 0 & \text{otherwise}, \end{array} \right. \quad \mathrm{H} = \left[ \begin{array}{cc} \mathrm{U} & -\mathrm{U} \\ -\mathrm{U} & \mathrm{U} \end{array} \right] \tag{4}$$

*It has first layer looks-linear structure if $\mathrm{H} = [\mathrm{U}; -\mathrm{U}]$.*

We impose this structure separately on the residual and skip branch but choose the corresponding submatrices wisely. To introduce RISOTTO, we only have to specify the corresponding submatrices for $\mathbf{W}_1^l$, $\mathbf{W}_2^l$, and $\mathbf{W}_{\text{skip}}^l$. The main idea of Risotto is to choose them so that the initial residual block acts as a linear orthogonal map.

The Type C residual block assumes that the skip connection is a projection such that $h_i^l(x) = \sum_{j \in N_l} \mathbf{W}_{ij,\text{skip}}^l * x_j^l$, where $\mathbf{W}_{\text{skip}}^l \in \mathbb{R}^{N_{l+1} \times N_l \times 1 \times 1}$ is a trainable convolutional tensor with kernel size $1 \times 1$. Thus, we can adapt the skip connections to compensate for the added activations of the residual branch as visualized in Fig. 3.

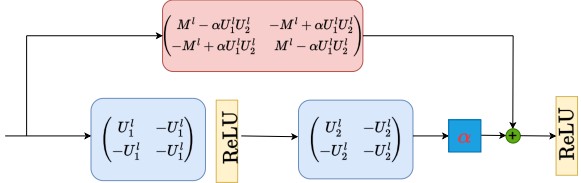

Figure 3: A Type C residual block is initialized with Risotto as visually explained.

**Definition 2.8** (RISOTTO for Type C residual blocks). *For a residual block of the form $\boldsymbol{x}^{l+1} = \phi(\alpha * f^l(\boldsymbol{x}^l) + h^l(\boldsymbol{x}^l))$, where $f^l(\boldsymbol{x}^l) = \mathbf{W}_2^l * \phi(\mathbf{W}_1^l * \boldsymbol{x}^l)$, $h^l(\boldsymbol{x}^l) = \mathbf{W}_{skip}^l * \boldsymbol{x}^l$, the weights $\mathbf{W}_1^l, \mathbf{W}_2^l$ and $\mathbf{W}_{skip}^l$ are initialized with looks-linear structure according to Def. 2.7 with the submatrices $U_1^l$, $U_2^l$ and $U_{skip}^l$ respectively. The matrices $U_1^l$, $U_2^l$, and $M^l$ be drawn independently and uniformly from all matrices with orthogonal rows or columns (depending on their dimension), while the skip submatrix is $U_{skip}^l = M^l - \alpha U_2^l U_1^l$.*

The Type B residual block poses the additional challenge that we cannot adjust the skip connections initially because they are defined by the identity and not trainable. Thus, we have to adapt the residual connections instead to compensate for the added input signal. To be able to distinguish the positive and the negative part of the input signal after the two convolutional layers, we have to pass it through the first ReLU without transformation and thus define $\mathbf{W}_1^l$ as identity mapping.

**Definition 2.9** (RISOTTO for Type B residual blocks). *For a residual block of the form $\boldsymbol{x}^{l+1} = \phi(\alpha * f^l(\boldsymbol{x}^l) + \boldsymbol{x}^l)$ where $f^l(\boldsymbol{x}^l) = \mathbf{W}_2^l * \phi(\mathbf{W}_1^l * \boldsymbol{x}^l)$, RISOTTO initializes the weight $\mathbf{W}_1^l$ as $w_{1,ijk_1'k_2'}^l = 1$ if $i = j, k_1' = \lfloor k_1/2 \rfloor, k_2' = \lfloor k_2/2 \rfloor$ and $w_{1,ijk_1'k_2'}^l = 0$ otherwise. $\mathbf{W}_2^l$ has looks-linear structure (according to Def. 2.7) with respect to a submatrix $U_2^l = M^l - (1/\alpha)\mathbb{I}$, where $M^l \in \mathbb{R}^{N_{l+1}/2 \times N_l/2}$ is a random matrix with orthogonal columns or rows, respectively.*

Note that particularly the type B initialization consists of a considerable amount of zero entries. To induce higher symmetry breaking and promote feature diversity Blumenfeld et al. (2020), we also study a noisy variant, N-RISOTTO, in which we add a small amount of noise $\epsilon\Sigma$ to our RISOTTO parameters, where $\epsilon = 10^{-4}$ and $\Sigma$ follows the He Normal distribution (He et al., 2015).

As we prove in the appendix, residual blocks initialized with RISOTTO preserve the norm of the input and cosine similarity of signals corresponding to different inputs not only on average but exactly, as exemplified by Fig. 2c (b). This addresses the drawbacks of initialization schemes that are based on independent weight entries, as discussed in the last section.

**Theorem 2.10** (RISOTTO **preserves signal norm and similarity**). *A residual block that is initialized with* RISOTTO *maps input activations $\boldsymbol{x}^l$ to output activations $\boldsymbol{x}^{l+1}$ so that the norm $||\boldsymbol{x}^{l+1}||^2 = ||\boldsymbol{x}^l||^2$ stays equal. The scalar product between activations corresponding to two inputs $\boldsymbol{x}$ and $\tilde{\boldsymbol{x}}$ are preserved in the sense that $\langle \hat{\boldsymbol{x}}_+^{l+1} - \hat{\boldsymbol{x}}_-^{l+1}, \tilde{\boldsymbol{x}}_+^{l+1} - \tilde{\boldsymbol{x}}_-^{l+1} \rangle = \langle \hat{\boldsymbol{x}}_+^l - \hat{\boldsymbol{x}}_-^l, \tilde{\boldsymbol{x}}_+^l - \tilde{\boldsymbol{x}}_-^l \rangle$.*

The full proof is presented Appendix A.3. It is straight forward, as the residual block is defined as orthogonal linear transform, which maintains distances of the sum of the separated positive and negative part of a signal. Like the residual block, the input-output Jacobian also is formed of orthogonal submatrices. It follows that RISOTTO induces perfect dynamical isometry for finite width and depth.

**Theorem 2.11** (RISOTTO **achieves exact dynamical isometry for residual blocks**). *A residual block whose weights are initialized with* RISOTTO *achieves exact dynamical isometry so that the singular values $\lambda \in \sigma(J)$ of the input-output Jacobian $J \in \mathbb{R}^{N_{l+1} \times k_{l+1} \times N_l \times k_l}$ fulfill $\lambda \in \{-1, 1\}$.*

The detailed proof is given in Appendix A.2. Since the weights are initialized so that the residual block acts as an orthogonal transform, also the input-output Jacobian is an isometry, which has the required spectral properties. Drawing on the well established theory of dynamical isometry (Chen et al., 2018; Saxe et al., 2013; Mishkin & Matas, 2015; Poole et al., 2016; Pennington et al., 2017), we therefore expect RISOTTO to enable fast and stable training of very deep ResNets, as we demonstrate next in experiments.

## 3  Experiments

In all our experiments, we use two kinds of ResNets consisting of residual blocks of Type B or Type C, as defined in Section 2.1 and visualized in Fig. 2a. ResNet (B) contain Type B residual blocks if the input and output dimension of the block is equal and a Type B block otherwise, but a ResNet (C) has Type C residual blocks throughout. All implementation details and our hyperparameter tuning approach are described in Appendix A.4. We use a learnable scalar $\alpha$ that is initialized as $\alpha = 1$ to balance skip and residual connections (see Fig. 2c (c)). Results for all cases are reported on the benchmark datasets CIFAR10, CIFAR100 (Krizhevsky et al., 2014), and Tiny ImageNet (Le & Yang, 2015). Additional key experiments on ImageNet are presented in Appendix A.9 that are in line with our findings on Tiny ImageNet.

Our main objective is to highlight three advantageous properties of our proposed initialization RISOTTO: (a) It enables stable and fast training of deep ResNets without any normalization methods and outperforms state-of-the-art schemes designed for this purpose (see Table 1 and Fig. 4 (center)). (b) Unnormalized it can compete with NF ResNets, an alternative normalization approach to BN, (see Fig. 4 (right) and Fig. 9). (c) It can outperform alternative initialization methods in combination with Batch Normalization (BN) (see Table 2 and Fig. 4 (left)). We hypothesize that this is enabled by the balance between skip and residual connections that is unique to RISOTTO.

**RISOTTO without BN** We start our empirical investigation by evaluating the performance of ResNets without any normalization layers. We compare our initialization scheme RISOTTO to the state-of-the-art baselines Fixup (Zhang et al., 2018) and SkipInit (De & Smith, 2020). Both these methods have been proposed as substitutes for BN and are designed to achieve the benefits of BN by scaling down weights with depth (Fixup) and biasing signal flow towards the skip connection (Fixup and SkipInit). Fixup has so far achieved the best performance for training ResNets without any form of normalization. We observe that as shown in Table 1, RISOTTO is able to outperform both Fixup and SkipInit. Moreover, we also observed in our experiments that Fixup and SkipInit are both susceptible to bad random seeds and can lead to many failed training runs. The unstable gradients at the beginning due to zero initialization of the last residual layer might be responsible for this phenomenon. RISOTTO produces stable results. With ResNet (C), it also achieves the overall highest accuracy for all three datasets. We have also evaluated N-Rissoto for ResNet50 (B) on **ImageNet** comparing with Fixup as shown in Fig. 10 in Appendix A.9. These results verify that a well balanced orthogonal initialization for residual blocks enables better training for different datasets and models of varying sizes.

| Dataset | ResNet | RISOTTO (ours) | Fixup | SkipInit |
|---|---|---|---|---|
| CIFAR10 | 18 (C) | **93.71 ± 0.11** | 92.05 ± 0.11 | 10.0 ± 0.03 |
| | 18 (B) | 92.1 ± 0.52 | 93.36 ± 0.15 | 92.57 ± 0.21 |
| CIFAR100 | 50 (C) | **60.63 ± 0.28** | 58.25 ± 1.64 | 1 ± 0 |
| | 50 (B) | 56.17 ± 0.39 | 59.26 ± 0.69 | 42.40 ± 0.67 |
| Tiny ImageNet | 50 (C) | **49.51 ± 0.06** | 48.07 ± 0.46 | 26.42 ± 9.7 |
| | 50 (B) | 47.02 ± 0.37 | 47.89 ± 0.45 | 31.57 ± 8.6 |

Table 1: **RISOTTO as a substitute for Batch Normalization** The mean test accuracy over 3 runs and 0.95 standard confidence intervals are reported to compare RISOTTO, Fixup, and SkipInit without using BN. RISOTTO is able to achieve the overall best results for each of the benchmark datasets.

**RISOTTO learns faster for deep ResNets.** Fig. 4 (center) for ResNet B consisting of 300 layers and Fig. 8b for ResNet101 on CIFAR100 demonstrate RISOTTO's ability to train deep networks without BN. While RISOTTO achieves a higher final performance for 300 layers and similar one as Fixup for 101 layers, RISOTTO trains faster in comparison. Also in combination with BN, RISOTTO can train faster, as Fig. 10 shows for ResNet50 (B) on ImageNet.

**Comparison with Normalization Free ResNets** NF ResNets (Brock et al., 2021a) that use weight standardization have been shown to outperform BN when used in combination with adaptive gradient clipping. We find that RISOTTO is able to outperform NF ResNets on Tiny ImageNet with He initialization He et al. (2016) as shown in Fig. 4 (right). Further experiments are reported in Fig. 9.

**RISOTTO in combination with BN** Despite its drawbacks, BN remains a popular method and is implemented per default, as it often leads to the best overall generalization performance for ResNets. Normalization free initialization schemes have been unable to compete with BN, even though Fixup has come close. Whether the performance of batch normalized networks can still be improved is therefore still a relevant question. Table 2 compares RISOTTO with the two variants of He initialization for normally distributed and uniformly distributed weights (He et al., 2016). We find that RISOTTO arrives at marginally lower performance on CIFAR10 but outperforms the standard methods on both CIFAR100 and Tiny ImageNet. On **ImageNet**, a single N-Risotto training run for ResNet50 (B) with BN obtains 65.67 test accuracy and He Uniform initialization 58.41 (see Figure 10 in Appendix A.9). These results and Table 1 showcase the versatility of RISOTTO, as it enables training without BN and can even improve training with BN (see also Fig. 4 (left)).

| Dataset | ResNet | RISOTTO (ours) | He Normal | He Uniform |
|---------|--------|----------------|-----------|------------|
| CIFAR10 | 18 (C) | $95.29 \pm 0.14$ | $\mathbf{95.38 \pm 0.15}$ | $95.32 \pm 0.07$ |
|         | 18 (B) | $94.93 \pm 0.07$ | $\underline{94.99 \pm 0.12}$ | $94.82 \pm 0.03$ |
| CIFAR100 | 50 (C) | $73.11 \pm 0.70$ | $76.17 \pm 0.25$ | $\underline{76.21 \pm 0.28}$ |
|          | 50 (B) | $\mathbf{78.45 \pm 0.08}$ | $77.40 \pm 0.25$ | $76.7 \pm 0.8$ |
| Tiny ImageNet | 50 (C) | $\mathbf{59.47 \pm 0.02}$ | $50.12 \pm 0.98$ | $53.91 \pm 0.59$ |
|               | 50 (B) | $\underline{58.73 \pm 0.29}$ | $52.21 \pm 2$ | $55.05 \pm 0.6$ |

Table 2: **RISOTTO with Batch Normalization** The mean test accuracy over 3 runs and 0.95 standard confidence intervals are reported to compare different initialization in combination with BN. RISOTTO outperforms the baseline methods with BN on CIFAR100 and Tiny ImageNet.

**Can we reduce the number of BN layers?** Considering the importance of BN on performance and how other methods struggle to compete with it, we note that reducing the number of BN layers in a network can still render similar benefits at reduced computational and memory costs. Often a single BN layer is sufficient and with Risotto, we are flexible in choosing its position. Fig. 4 (right) reports results for BN after the last layer, which usually achieves the best performance. Additional figures in the appendix (Fig. 5-7) explore optional BN layer placements in more detail.

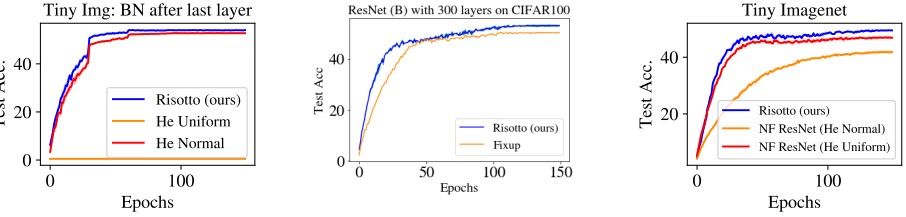

Figure 4: (left) A single BN layer is placed after the last residual block of a ResNet50 (C) on Tiny Imagenet. (center) A 300 layer ResNet (B) with RISOTTO versus Fixup. (right) RISOTTO versus NF ResNets (C) on Tiny Imagenet.

## 4 CONCLUSIONS

We have introduced a new initialization method for residual networks with ReLU activations, RISOTTO. It enables residual networks of any depth and width to achieve exact dynamical isometry at initialization. Furthermore, it can balance the signal contribution from the residual and skip branches instead of suppressing the residual branch initially. This does not only lead to higher feature diversity but also promotes stable and faster training. In practice that is highly effective, as we demonstrate for multiple standard benchmark datasets. We show that RISOTTO competes with and often outperforms Batch Normalization free methods and even improves the performance of ResNets that are trained with Batch Normalization. While we have focused our exposition on ResNets with ReLU activation functions, in future, RISOTTO could also be transferred to transformer architectures similarly to the approach by Bachlechner et al. (2021). In addition, the integration into the ISONet (Qi et al., 2020) framework could potentially lead to further improvements.

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

# A  APPENDIX

## A.1  SIGNAL PROPAGATION

Recall our definition of a residual block in Equation (1):

$$\boldsymbol{z}^0 := \mathbf{W}^0 \boldsymbol{x}, \quad \boldsymbol{x}^l = \phi(\boldsymbol{z}^{l-1}), \quad \boldsymbol{z}^l := \alpha_l f^l(\boldsymbol{x}^l) + \beta_l h^l(\boldsymbol{x}^l), \quad \boldsymbol{z}^{out} := \mathbf{W}^{out} P(\boldsymbol{x}^L) \quad (5)$$

for $1 \leq l \leq L$ with $f^l(\boldsymbol{x}^l) = \mathbf{W}_2^l \phi(\mathbf{W}_1^l \boldsymbol{x}^l)$, $h^l(\boldsymbol{x}^l) = \mathbf{W}_{\text{skip}}^l \boldsymbol{x}^l$, where the biases have been set to zero. In the following theoretical derivations we focus on fully-connected networks for simplicity so that $\mathbf{W}_2^l \in \mathbb{R}^{N_{l+1} \times N_{m_l}}$, $\mathbf{W}_1^l \in \mathbb{R}^{N_{m_l} \times N_l}$, and $\mathbf{W}_{\text{skip}}^l \in \mathbb{R}^{N_{l+1} \times N_l}$. The general principle could also be transferred to convolutional layers similarly to the mean field analysis by Xiao et al. (2018).

A common choice for the initialization of the parameters is defined as follows.

**Definition A.1** (Normal ResNet Initialization for fullly-connected residual blocks). *Let a neural network consist of fully-connected residual blocks as defined by Equ. (5). All biases are initialized as $0$ and all weight matrix entries are independently normally distributed with $w_{ij,2}^l \sim \mathcal{N}\left(0, \sigma_{l,2}^2\right)$, $w_{ij,1}^l \sim \mathcal{N}\left(0, \sigma_{l,1}^2\right)$, and $w_{ij,skip}^l \sim \mathcal{N}\left(0, \sigma_{l,skip}^2\right)$. Then the Normal ResNet Initialization is defined by the choice $\sigma_{l,1} = \sqrt{\frac{2}{N_{m_l}}}$, $\sigma_{l,2} = \sqrt{\frac{2}{N_{l+1}}}$, $\sigma_{l,skip} = \sqrt{\frac{2}{N_{l+1}}}$, and $\alpha_l, \beta_l \geq 0$ that fulfill $\alpha_l^2 + \beta_l^2 = 1$.*

Our objective is to analyze the distribution of the signals $\boldsymbol{x}^{l+1}$ and $\tilde{\boldsymbol{x}}^{l+1}$, which correspond to the random neuron state of an initial neural network that is evaluated in input $\boldsymbol{x}^0$ or $\tilde{\boldsymbol{x}}^0$, respectively. More precisely, we derive the average squared signal norm and the covariance of two signals that are evaluated in different inputs.

We start with the squared signal norm and, for convenience, restate Theorem 2.10 before the proof.

**Theorem A.2** (Theorem 2.10 in main manuscript). *Let a neural network consist of residual blocks as defined by Equ. (1) or Equ. (5) that start with a fully-connected layer at the beginning $\mathbf{W}^0$, which contains $N_1$ output channels. Assume that all biases are initialized as $0$ and that all weight matrix entries are independently normally distributed with $w_{ij,2}^l \sim \mathcal{N}\left(0, \sigma_{l,2}^2\right)$, $w_{ij,1}^l \sim \mathcal{N}\left(0, \sigma_{l,1}^2\right)$, and $w_{ij,skip}^l \sim \mathcal{N}\left(0, \sigma_{l,skip}^2\right)$. Then the expected squared norm of the output after one fully-connected layer and $L$ residual blocks applied to input $x$ is given by*

$$\mathbb{E}\left(\|\boldsymbol{x}^L\|^2\right) = \frac{N_1}{2}\sigma_0^2 \prod_{l=1}^{L-1} \frac{N_{l+1}}{2}\left(\alpha_l^2 \sigma_{l,2}^2 \sigma_{l,1}^2 \frac{N_{m_l}}{2} + \beta_l^2 \sigma_{l,skip}^2\right) \|\boldsymbol{x}\|^2.$$

*Proof.* First, we study how the signal is transformed by a single layer before we deduce successively the final output signal. To do so, we assume that the signal of the previous layer is given. This means we condition the expectation on the parameters of the previous layers and thus $\boldsymbol{x}^l$ and $\tilde{\boldsymbol{x}}^l$. For notational convenience, we define $\mathbb{E}_l(\boldsymbol{z}) = \mathbb{E}\left(\boldsymbol{z} \mid \boldsymbol{x}^l, \tilde{\boldsymbol{x}}^l\right)$ and skip the index $l$ in the following derivations. We write: $\boldsymbol{x} = \boldsymbol{x}^{l+1}$, $\underline{\boldsymbol{x}} = \boldsymbol{x}^l$, $\boldsymbol{z} = \boldsymbol{z}^l$, $f(\underline{\boldsymbol{x}}) = f^l(\underline{\boldsymbol{x}}) = \mathbf{W}_2^l \phi(\mathbf{W}_1^l \underline{\boldsymbol{x}})$, $h(\underline{\boldsymbol{x}}) = h^l(\underline{\boldsymbol{x}}) = \mathbf{W}_{\text{skip}}^l \underline{\boldsymbol{x}}$, $\alpha = \alpha_l$, $\beta = \beta_l$. Given all parameters from the previous layers, we deduce

$$\mathbb{E}\left(\|\boldsymbol{x}\|^2 \mid \underline{\boldsymbol{x}}\right) = \sum_{i=1}^{N_{l+1}} \mathbb{E}\left((x_i)^2\right) = N_{l+1}\mathbb{E}\left((x_1)^2\right) = \frac{N_{l+1}}{2}\mathbb{E}_l\left(z_1\right)^2 \quad (6)$$

The first equality follows from the fact that all random parameters are independent and the signal components are identically distributed. The third equality holds because the distribution of each signal component is symmetric around zero so that the ReLU projects half of the signal away but

the contribution to the average of the squared signal is just cut in half. We continue with

$$\mathbb{E}_l \left(z_1\right)^2 = \mathbb{E}_l \left(\alpha \sum_{i=1}^{N_{m_l}} w_{2,1i}\phi\left(\sum_{j=1}^{N_l} w_{1,ij}\underline{x}_j\right) + \beta \sum_{k=1}^{N_l} w_{\text{skip},1k}\underline{x}_k\right)^2 \tag{7}$$

$$= \alpha^2 \sum_{i=1}^{N_{m_l}} \mathbb{E}_l\left(w_{2,ij}^2\right)\mathbb{E}_l\left(\left\|\phi\left(\sum_{j=1}^{N_l} w_{1,ij}\underline{x}_j\right)\right\|^2\right) + \beta^2 \sum_{k=1}^{N_l} \mathbb{E}_l\left(w_{\text{skip},1k}^2\right)\underline{x}_k^2 \tag{8}$$

$$= \alpha^2 \sigma_{l,2}^2 N_{m_l} \mathbb{E}_l\left(\left\|\phi\left(\sum_{j=1}^{N_l} w_{1,1j}\underline{x}_j\right)\right\|^2\right) + \beta^2 \sigma_{l,\text{skip}}^2 \|\underline{x}\|^2 \tag{9}$$

$$= \alpha^2 \sigma_{l,2}^2 N_{m_l} \frac{1}{2}\mathbb{E}_l\left(\sum_{j=1}^{N_l} w_{1,1j}\underline{x}_j\right)^2 + \beta^2 \sigma_{l,\text{skip}}^2 \|\underline{x}\|^2 \tag{10}$$

$$= \left(\alpha^2 \sigma_{l,2}^2 \sigma_{l,1}^2 N_{m_l} \frac{1}{2} + \beta^2 \sigma_{l,\text{skip}}^2 N_{l+1}\right)\|\underline{x}\|^2, \tag{11}$$

as all weight entries are independent and the expectation is a linear operation. To obtain Equation (10), we just repeated the same argument as for Equation 6 to take care of the ReLU. Afterwards, we used again the independence of the weights $w_{1,1j}$.

From repeated evaluation of Equations (6) and (11), we obtain

$$\mathbb{E}\left(\left\|\boldsymbol{x}^L\right\|^2\right) = \frac{N_1}{2}\sigma_0^2 \prod_{l=1}^{L-1} \frac{N_{l+1}}{2}\left(\alpha^2 \sigma_{l,2}^2 \sigma_{l,1}^2 N_{m_l} \frac{1}{2} + \beta^2 \sigma_{l,\text{skip}}^2\right)\|\boldsymbol{x}\|^2 \tag{12}$$

for $\boldsymbol{x} = \boldsymbol{x}^0$.  $\qquad\qquad\square$

To make sure that the signal norm neither explodes or vanishes for very deep networks, we would need to choose the weight variances so that $\frac{N_1}{2}\sigma_0^2 \prod_{l=1}^{L} \frac{N_{l+1}}{2}\left(\alpha^2 \sigma_{l,2}^2 \sigma_{l,1}^2 N_{m_l} \frac{1}{2} + \beta^2 \sigma_{l,\text{skip}}^2\right) \approx 1$. In the common normal ResNet initialization, this is actually achieved, since $\sigma_{l,1} = \sqrt{2/N_{m_l}}$, $\sigma_{l,2} = \sigma_{l,\text{skip}} = \sqrt{2/N_{l+1}}$, $\sigma_0 = \sqrt{2/N_1}$, and $\alpha^2 + \beta^2 = 1$ even preserve the average norm in every layer.

How does this choice affect whether signals for different inputs are distinguishable? To answer this question, we analyze the covariance of the neuron state for two different inputs. We begin again with analyzing the transformation of a single layer and condition on all parameters of the previous layers. To obtain a lower bound on the covariance, the following Lemma will be helpful. It has been derived by Burkholz & Dubatovka (2019) for Theorem 5.

**Lemma A.3.** *Assume that two random variables $z_1$ and $z_2$ are jointly normally distributed as $z \sim \mathcal{N}(0, V)$ with covariance matrix $V$. Then, the covariance of the ReLU transformed variables $x_1 = \phi(z_1)$ and $x_2 = \phi(z_2)$ is*

$$\mathbb{E}\left(x_1 x_2\right) = \sqrt{v_{11}v_{22}}\left(g(\rho)\rho + \frac{\sqrt{1-\rho^2}}{2\pi}\right) \tag{13}$$

$$\geq \sqrt{v_{11}v_{22}}\left(\frac{1}{4}(\rho + 1) - \tilde{c}\right) = \frac{1}{4}v_{12} + c\sqrt{v_{11}v_{22}}, \tag{14}$$

*where $\rho = v_{11}v_{22}/v_{12}$ and $g(\rho)$ is defined as $g(\rho) = \frac{1}{\sqrt{2\pi}}\int_0^\infty \Phi\left(\frac{\rho}{\sqrt{1-\rho^2}}u\right)\exp\left(-\frac{1}{2}u^2\right)\,du$ for $|\rho| \neq 1$ and $g(-1) = 0$, $g(1) = 0.5$. The constant fulfills $0.24 \leq c \leq 0.25$.*

In the following, we assume that all weight parameters are normally distributed so that we can use the above lemma. However, other parameter distributions in large networks would also lead to similar results, as the central limit theorem implies that the relevant quantities are approximately normally distributed.

We study the covariance of the signals $\boldsymbol{x}^{l+1} = \boldsymbol{x}$ and $\tilde{\boldsymbol{x}}^{l+1} = \tilde{\boldsymbol{x}}$, which correspond to the random neuron state of an initial neural network that is evaluated in input $\boldsymbol{x}^0$ or $\tilde{\boldsymbol{x}}^0$, respectively.

**Theorem A.4** (Theorem 2.5 in main manuscript). *Let a fully-connected residual block be given as defined by Equ. (1) or Equ. (5). Assume that all biases are initialized as $0$ and that all weight matrix entries are independently normally distributed with $w_{ij,2}^l \sim \mathcal{N}\left(0, \sigma_{l,2}^2\right)$, $w_{ij,1}^l \sim \mathcal{N}\left(0, \sigma_{l,1}^2\right)$, and $w_{ij,skip}^l \sim \mathcal{N}\left(0, \sigma_{l,skip}^2\right)$. Let $x^{l+1}$ denote the neuron states of Layer $l+1$ for input $\boldsymbol{x}$ and $\tilde{\boldsymbol{x}}^{l+1}$ the same neurons but for input $\tilde{\boldsymbol{x}}$. Then their covariance given all parameters of the previous layers is given as $\mathbb{E}_l\left(\langle \boldsymbol{x}^{l+1}, \tilde{\boldsymbol{x}}^{l+1}\rangle\right)$*

$$\geq \frac{1}{4}\frac{N_{l+1}}{2}\left(\alpha_l^2\sigma_{l,2}^2\sigma_{l,1}^2\frac{N_{m_l}}{2} + 2\beta_l^2\sigma_{l,skip}^2\right)\langle\boldsymbol{x}^l, \tilde{\boldsymbol{x}}^l\rangle + \frac{c}{4}\alpha_l^2 N_{l+1}\sigma_{l,2}^2\sigma_{l,1}^2 N_{m_l}\left\|\boldsymbol{x}^l\right\|\left\|\tilde{\boldsymbol{x}}^l\right\| \quad (15)$$

$$+ \mathbb{E}_{\mathbf{W}_1^l}\left(\sqrt{\left(\alpha_l^2\sigma_{l,2}^2\left\|\phi(\mathbf{W}_1^l\boldsymbol{x}^l)\right\|^2 + \beta_l^2\sigma_{l,skip}^2\left\|\boldsymbol{x}^l\right\|^2\right)\left(\alpha_l^2\sigma_{l,2}^2\left\|\phi(\mathbf{W}_1^l\tilde{\boldsymbol{x}}^l)\right\|^2 + \beta_l^2\sigma_{l,skip}^2\left\|\tilde{\boldsymbol{x}}^l\right\|^2\right)}\right),$$

*where the expectation $\mathbb{E}_l$ is taken with respect to the initial parameters $\mathbf{W}_2^l$, $\tilde{\mathbf{W}}_1^l$, and $\mathbf{W}_{skip}^l$.*

*Proof.* Let us assume again that all parameters of the previous layers are given in addition to the parameters of the first residual layer $\mathbf{W}_1$ and use our notation from the proof of Theorem 2.10.

Based on similar arguments that we used for the derivation of average squared signal norm, we observe that $\boldsymbol{z} = \boldsymbol{z}^l$ and $\tilde{\boldsymbol{z}} = \tilde{\boldsymbol{z}}^l$ are jointly normally distributed. In particular, the components $z_i$ are identically distributed for the same input. The same component $z_i$ and $\tilde{z}_i$ for different inputs has covariance matrix $\mathbf{V}$ with entries $v_{11} = \alpha^2\sigma_{l,2}^2\left\|\phi(\mathbf{W}_1\underline{\boldsymbol{x}})\right\|^2 + \beta^2\sigma_{l,skip}^2\left\|\underline{\boldsymbol{x}}\right\|^2$, $v_{22} = \alpha^2\sigma_{l,2}^2\left\|\phi(\mathbf{W}_1\tilde{\underline{\boldsymbol{x}}})\right\|^2 + \beta^2\sigma_{l,skip}^2\left\|\tilde{\underline{\boldsymbol{x}}}\right\|^2$, and $v_{12} = \alpha^2\sigma_{l,2}^2\langle\phi(\mathbf{W}_1\underline{\boldsymbol{x}}), \phi(\mathbf{W}_1\tilde{\underline{\boldsymbol{x}}})\rangle + \beta^2\sigma_{l,skip}^2\langle\underline{\boldsymbol{x}}, \tilde{\underline{\boldsymbol{x}}}\rangle$.

$$\mathbb{E}_l\left(\langle\boldsymbol{x}, \tilde{\boldsymbol{x}}\rangle\right) = \mathbb{E}_{\mathbf{W}_1}\mathbb{E}_l\left(\langle\boldsymbol{x}, \tilde{\boldsymbol{x}}\rangle \mid \mathbf{W}_1\right) = \sum_{i=1}^{N_{l+1}}\mathbb{E}_{\mathbf{W}_1}\mathbb{E}_l\left(\phi(z_i)\phi(\tilde{z}_i) \mid \mathbf{W}_1\right) \quad (16)$$

$$\geq N_{l+1}\mathbb{E}_{\mathbf{W}_1}\left(\frac{1}{4}v_{12} + c\sqrt{v_{11}v_{22}}\right) = N_{l+1}\mathbb{E}_{\mathbf{W}_1}\left(\frac{1}{4}v_{12} + c\sqrt{v_{11}v_{22}}\right) \quad (17)$$

where we applied Lemma A.3 to obtain the inequality and used the fact that the entries of the variance matrix are identically distributed for different $(i)$ with respect to $\mathbf{W}_1$.

We can compute the first term $\mathbb{E}_{\mathbf{W}_1}v_{12}$ by using Lemma A.3 again, as $\mathbf{W}_1\underline{\boldsymbol{x}}$ and $\mathbf{W}_1\tilde{\underline{\boldsymbol{x}}}$ are jointly normally distributed given $\underline{\boldsymbol{x}}$ and $\tilde{\underline{\boldsymbol{x}}}$. The associated covariance matrix $S$ for one component corresponding to two different inputs has entries $s_{11} = \sigma_{l,1}^2\left\|\underline{\boldsymbol{x}}\right\|^2$, $s_{22} = \sigma_{l,1}^2\left\|\tilde{\underline{\boldsymbol{x}}}\right\|^2$, and $s_{12} = \sigma_{l,1}^2\langle\underline{\boldsymbol{x}}, \tilde{\underline{\boldsymbol{x}}}\rangle$. Lemma A.3 gives us therefore

$$\mathbb{E}_{\mathbf{W}_1}\left(v_{12}\right) = \mathbb{E}_{\mathbf{W}_1}\left(\alpha^2\sigma_{l,2}^2\langle\phi(\mathbf{W}_1\underline{\boldsymbol{x}}), \phi(\mathbf{W}_1\tilde{\underline{\boldsymbol{x}}})\rangle + \beta^2\sigma_{l,skip}^2\langle\underline{\boldsymbol{x}}, \tilde{\underline{\boldsymbol{x}}}\rangle\right) \quad (18)$$

$$\geq \alpha^2\sigma_{l,2}^2 N_{m_l}\left(\frac{1}{4}\sigma_{l,1}^2\langle\underline{\boldsymbol{x}}, \tilde{\underline{\boldsymbol{x}}}\rangle + c\sigma_{l,1}^2\left\|\underline{\boldsymbol{x}}\right\|\left\|\tilde{\underline{\boldsymbol{x}}}\right\|\right) + \beta^2\sigma_{l,skip}^2\langle\underline{\boldsymbol{x}}, \tilde{\underline{\boldsymbol{x}}}\rangle. \quad (19)$$

Determining the second part of Equation (17) is more involved:

$$\mathbb{E}_{\mathbf{W}_1}\left(\sqrt{v_{11}v_{22}}\right)$$

$$= \mathbb{E}_{\mathbf{W}_1}\left(\sqrt{\left(\alpha^2\sigma_{l,2}^2\|\phi(\mathbf{W}_1\boldsymbol{x})\|^2 + \beta^2\sigma_{l,\text{skip}}^2\|\boldsymbol{x}\|^2\right)\left(\alpha^2\sigma_{l,2}^2\|\phi(\mathbf{W}_1\tilde{\boldsymbol{x}})\|^2 + \beta^2\sigma_{l,\text{skip}}^2\|\tilde{\boldsymbol{x}}\|^2\right)}\right) \tag{20}$$

$$= \beta^2\sigma_{l,\text{skip}}^2\|\boldsymbol{x}\|\,\|\tilde{\boldsymbol{x}}\|\,\mathbb{E}_{\mathbf{W}_1}\sqrt{\left(\gamma\left\|\phi\left(\mathbf{W}_1\frac{\boldsymbol{x}}{\|\boldsymbol{x}\|}\right)\right\|^2 + 1\right)\left(\gamma\left\|\phi\left(\mathbf{W}_1\frac{\tilde{\boldsymbol{x}}}{\|\tilde{\boldsymbol{x}}\|}\right)\right\|^2 + 1\right)} \tag{21}$$

$$\geq \beta^2\sigma_{l,\text{skip}}^2\|\boldsymbol{x}\|\,\|\tilde{\boldsymbol{x}}\|\,\mathbb{E}_{\mathbf{W}_1}\sqrt{\left(\gamma\left\|\phi\left(\mathbf{W}_1\frac{\boldsymbol{x}}{\|\boldsymbol{x}\|}\right)\right\|^2 + 1\right)\left(\gamma\left\|\phi\left(-\mathbf{W}_1\frac{\boldsymbol{x}}{\|\boldsymbol{x}\|}\right)\right\|^2 + 1\right)} \tag{22}$$

$$\geq \beta^2\sigma_{l,\text{skip}}^2\|\boldsymbol{x}\|\,\|\tilde{\boldsymbol{x}}\|\,\mathbb{E}_{\mathbf{W}_1}\sqrt{\left(\gamma\sum_{j=1}^{M}w_{1,j1}^2 + 1\right)}\sqrt{\left(\gamma\sum_{j=M+1}^{N_{m_l}}w_{1,j1}^2 + 1\right)} \tag{23}$$

$$= \beta^2\sigma_{l,\text{skip}}^2\|\boldsymbol{x}\|\,\|\tilde{\boldsymbol{x}}\|\,\mathbb{E}_{\mathbf{M}}\mathbb{E}_{\mathbf{Y}}\left[\sqrt{\left(\gamma\sigma_{l,1}^2\frac{N_{m_l}}{2}\left(\frac{2}{N_{m_l}}\sum_{j=1}^{M}y_j^2\right) + 1\right)}\right.$$

$$\left.\times\sqrt{\left(\gamma\sigma_{l,1}^2\frac{N_{m_l}}{2}\left(\frac{2}{N_{m_l}}\sum_{j=M+1}^{N_{m_l}}y_j^2\right) + 1\right)}\right]$$

$$\approx \beta^2\sigma_{l,\text{skip}}^2\|\boldsymbol{x}\|\,\|\tilde{\boldsymbol{x}}\|\left(\gamma\sigma_{l,1}^2\frac{N_{m_l}}{2} + 1\right) = \|\boldsymbol{x}\|\,\|\tilde{\boldsymbol{x}}\|\left(\alpha^2\sigma_{l,2}^2\sigma_{l,1}^2\frac{N_{m_l}}{2} + \beta^2\sigma_{l,\text{skip}}^2\right) \tag{24}$$

In Equation (22), we have used the fact that $\mathbb{E}_{\mathbf{W}_1}\left(\sqrt{v_{11}v_{22}}\right)$ is monotonically increasing in the covariance $s_{12}$. Thus, the minimum is attained for perfectly negative associations between $\boldsymbol{x}$ and $\tilde{\boldsymbol{x}}$ and thus $\tilde{\boldsymbol{x}} = -\boldsymbol{x}$. To simplify the derivation, we further study the case $\boldsymbol{x} = (1,0,0,0,...)^T$. It follows that either $\phi(w_{1,j1})$ or $\phi(-w_{1,j1})$ is positive while the other one is zero. To ease the notation, by reindexing, we can assume that the first $M$ components fulfill $\phi(w_{1,j1}) > 0$, while the remaining $N_{m_l} - M$ components fulfill the opposite. Note that because $w_{1,j1}$ is distributed symmetrically around zero, $M \sim \text{Bin}(N_{m_l}, 0.5)$ is a binomially distributed random variable with success probability 0.5. Thus, $\mathbb{E}M = N_{m_l}/2$. To make the dependence on $N_{m_l}$ of the different variables more obvious, we have replaced the random variables $w_{1,j1}$ that are normally distributed with standard deviation $\sigma_1$ by standard normally distributed random variables $y_j$ with standard deviation 1. This makes the use of the law of large numbers in the last equation more apparent. Note that this approximation is only accurate for large $N_{m_l} >> 1$, which is usually fulfilled in practice.

Finally, combining Equations (17), (19), and (20), we receive

$$\mathbb{E}_l\left(\langle\boldsymbol{x},\tilde{\boldsymbol{x}}\rangle\right) \geq \frac{1}{16}\alpha^2 N_{l+1}\sigma_{l,2}^2\sigma_{l,1}^2 N_{m_l}\langle\boldsymbol{x},\tilde{\boldsymbol{x}}\rangle + \frac{c}{4}\alpha^2 N_{l+1}\sigma_{l,2}^2\sigma_{l,1}^2 N_{m_l}\|\boldsymbol{x}\|\,\|\tilde{\boldsymbol{x}}\| + \frac{1}{4}\beta^2 N_{l+1}\sigma_{l,\text{skip}}^2\langle\boldsymbol{x},\tilde{\boldsymbol{x}}\rangle$$

$$+ \mathbb{E}_{\mathbf{W}_1}\left(\sqrt{\left(\alpha^2\sigma_{l,2}^2\|\phi(\mathbf{W}_1\boldsymbol{x})\|^2 + \beta^2\sigma_{l,\text{skip}}^2\|\boldsymbol{x}\|^2\right)\left(\alpha^2\sigma_{l,2}^2\|\phi(\mathbf{W}_1\tilde{\boldsymbol{x}})\|^2 + \beta^2\sigma_{l,\text{skip}}^2\|\tilde{\boldsymbol{x}}\|^2\right)}\right) \tag{25}$$

$$\approx \frac{N_{l+1}}{2}\left(\frac{\alpha^2}{4}\frac{\sigma_{l,1}^2 N_{m_l}}{2}\sigma_{l,2}^2 + \frac{\beta^2}{2}\sigma_{l,\text{skip}}^2\right)\langle\boldsymbol{x},\tilde{\boldsymbol{x}}\rangle$$

$$+ c\frac{N_{l+1}}{2}\left(\alpha^2\frac{\sigma_{l,1}^2 N_{m_l}}{2}\sigma_{l,2}^2 + \left(2\alpha^2\sigma_{l,2}^2\sigma_{l,1}^2\frac{N_{m_l}}{2} + 2\beta^2\sigma_{l,\text{skip}}^2\right)\right)\|\boldsymbol{x}\|\,\|\tilde{\boldsymbol{x}}\| \tag{26}$$

$$\square$$

To understand the problem that these derivations imply, we next choose the weight parameters so that the squared norm signal is preserved from one layer to the next and, for simplicity, study the case in which $\|\boldsymbol{x}\| = \|\tilde{\boldsymbol{x}}\| = 1$. Then we have

$$\mathbb{E}_l\left(\langle \boldsymbol{x}, \tilde{\boldsymbol{x}} \rangle\right) \geq \frac{1+\beta^2}{4} \langle \underline{\boldsymbol{x}}, \underline{\tilde{\boldsymbol{x}}} \rangle + c(\alpha^2 + 2) \approx \frac{1+\beta^2}{4} \langle \underline{\boldsymbol{x}}, \tilde{\boldsymbol{x}} \rangle + \frac{\alpha^2}{4} + \frac{1}{2}. \tag{27}$$

Thus, the similarity of signals corresponding to different inputs increases always by at least a constant amount on average. Repeating the above bound layerwise, at Layer $L$ we receive for $\gamma_1 = \frac{1+\beta^2}{4} \leq \frac{1}{2}$ and $\gamma_2 = c(\alpha^2 + 2)$:

$$\mathbb{E}\left(\langle \boldsymbol{x}^L, \tilde{\boldsymbol{x}}^L \rangle\right) \geq \gamma_1^L \langle \boldsymbol{x}, \tilde{\boldsymbol{x}} \rangle + \gamma_2 \sum_{k=0}^{L-1} \gamma_1^k = \frac{\gamma_2}{1 - \gamma_1}\left(1 - \gamma_1^L\right). \tag{28}$$

According to our bound, output signals of very deep networks become more similar with increasing depth until they are almost indistinguishable. This phenomenon poses a great challenge for the trainability of deep resdidual neural networks with standard initialization schemes. Note that the case without skip-connections is also covered by the choice $\alpha = 1$ and $\beta = 0$. Interestingly, nonzero skip-connections ($\beta > 0$) fight the increasing signal similarity by giving more weight to the original signal similarity (increased $\gamma_1$) while decreasing the constant contribution of $\gamma_2$. This enables training of deeper models but cannot solve the general problem that increasingly deep models become worse in distinguishing different inputs initially. Even the best case scenario of $\alpha = 0$ and $\beta = 1$ leads eventually to forgetting of the original input association, since $\gamma_1 = 0.5 < 1$. With $\gamma_2 \approx 0.5$, the overall signal similarity $\mathbb{E}\left(\langle \boldsymbol{x}^L, \tilde{\boldsymbol{x}}^L \rangle\right)$ converges to 1 for $L \to \infty$ irrespective of the input similarity. Thus, every input signal is essentially mapped to the same vector for very deep networks, which explains the following insight.

**Insight A.5** (Insight 2.6 in main paper). *Let a fully-connected ResNet be given whose parameters are drawn according to Definition A.1. It follows from Theorem 2.5 that the outputs corresponding to different inputs become more difficult to distinguish for increasing depth L. In the mean field limit $N_{m_l} \to \infty$, the covariance of the signals is lower bounded by*

$$\mathbb{E}\left(\langle \boldsymbol{x}^L, \tilde{\boldsymbol{x}}^L \rangle\right) \geq \gamma_1^L \langle \boldsymbol{x}, \tilde{\boldsymbol{x}} \rangle + \gamma_2 \sum_{k=0}^{L-1} \gamma_1^k = \gamma_1^L \langle \boldsymbol{x}, \tilde{\boldsymbol{x}} \rangle + \frac{\gamma_2}{1 - \gamma_1}\left(1 - \gamma_1^L\right) \tag{29}$$

*for $\gamma_1 = \frac{1+\beta^2}{4} \leq \frac{1}{2}$ and $\gamma_2 = c(\alpha^2 + 2)$ and $E_{l-1}\|\boldsymbol{x}^l\|\|\tilde{\boldsymbol{x}}^l\| \approx 1$.*

However, our orthogonal initialization scheme Risotto does not suffer from increasing similarity of outputs corresponding to different inputs.

## A.2 DYNAMICAL ISOMETRY INDUCED BY RISOTTO

**Theorem A.6** (Theorem 2.11 in the main paper). *A residual block (of type B or type C) whose weights are initialized with RISOTTO achieves exact dynamical isometry so that the singular values $\lambda \in \sigma(J)$ of the input-output Jacobian $J \in \mathbb{R}^{N_{l+1} \times k_{l+1} \times N_l \times k_l}$ fulfill $\lambda \in \{-1, 1\}$.*

*Proof.* Consider a single element of the output activation of a type C residual block at layer $l$. At initialization, an output activation component at layer $l$ is

$$x_{ik}^{l+1} = \alpha \sum_{n \in N_{m_l}} w_{ink_1/2k_2/2,2}^l \sum_{m \in N_l} \phi(w_{nmk_1/2k_2/2,1}^l x_{mk}^l) + \sum_{n \in N_l} w_{in,\text{skip}}^l x_{nk}^l \tag{30}$$

since RISOTTO initializes only the central elements of every 2-D filter to nonzero values and all the other values are zero which reduces the convolution to a simple summation. For the subsequent calculations we ignore the filter dimension indices in the weights. Now the positive part of the output

$x_{ik}^{l+1}$ where $i \in [0, N_{l+1}/2]$ is given by

$$x_{ik}^{l+1} = \alpha \sum_{n \in N_{m_l}} w_{in,2}^l \sum_{j \in N_l} \phi(w_{nj,1}^l x_{jk}^l) + \sum_{n \in N_l} w_{in,\text{skip}}^l x_{nk}^l \tag{31}$$

$$= \alpha \sum_{n \in N_{m_l}/2} u_{in,2}^l \sum_{j \in N_l/2} \phi(u_{nj,1}^l x_{jk}^l) - \phi(-u_{nj,1}^l x_{jk}^l) + \sum_{n \in N_l} w_{in,\text{skip}}^l x_{nk}^l \tag{32}$$

$$= \alpha \sum_{n \in N_{m_l}/2} \sum_{j \in N_l/2} u_{in,2}^l u_{nj,1}^l x_{jk}^l + \sum_{n \in N_l/2} \left( m_{in}^l - \alpha \sum_{n \in N_{m_l}/2} u_{in,2}^l u_{nj,1}^l \right) x_{nk}^l \tag{33}$$

$$= \sum_{n \in N_l/2} m_{in}^l x_{nk}^l \tag{34}$$

Taking the derivative of the output $x_{ik}^{l+1}$ wrt an input element $x_{nk}^l$.

$$\frac{\partial x_{ik}^{l+1}}{\partial x_{nk}^l} = m_{in}^l \tag{35}$$

We can now obtain the input output Jacobian for $i \in [0, N_{l+1}/2]$ and $j \in [0, N_l/2]$ as

$$[J]_{ik'_{l+1}jk'_l}^l = \begin{cases} m_{ij}^l & \text{if } k'_{l+1} = k'_l \\ 0 & \text{otherwise} \end{cases} \tag{36}$$

And since $\mathbf{M}^l$ is orthogonal, the singular values of $J^l$ are one across the dimensions $i, j$. Due to the looks linear form of the input at the weights the complete Jacobian also takes the looks linear form as

$$[J]_{ik'_{l+1}jk'_l}^l = \begin{cases} m_{ij}^l & \text{if } k'_{l+1} = k'_l, i \in [0, N_{l+1}/2], j \in [0, N_l/2] \\ -m_{ij}^l & \text{if } k'_{l+1} = k'_l, i \in [N_{l+1}/2, N_{l+1}], j \in [0, N_l/2] \\ -m_{ij}^l & \text{if } k'_{l+1} = k'_l, i \in [0, N_{l+1}/2], j \in [N_l/2, N_l] \\ m_{ij}^l & \text{if } k'_{l+1} = k'_l, i \in [N_{l+1}/2, N_{l+1}], j \in [N_l/2, N_l] \\ 0 & \text{otherwise} \end{cases} \tag{37}$$

The same argument follows for type B residual blocks. Hence, since $\mathbf{M}^l$ is an orthogonal matrix, the singular values of the input output Jacobian of the residual blocks initialized with RISOTTO are exactly $\{1, -1\}$ for all depths and width and not just in the infinite width limit.

## A.3 SIGNAL PROPAGATION WITH RISOTTO

Now we closely analyze how RISOTTO transforms the input signal for residual blocks when initialized using Definitions 2.9 and 2.8. RISOTTO creates effectively an orthogonal mapping that induces DI. Note that our analysis of convolutional tensors is simplified to evaluating matrix operations since the Delta Orthogonal initialization simplifies a convolution to an effective matrix multiplication in the channel dimension of the input. Specifically, we can track the changes in the submatrices used in the initialization of a residual block with RISOTTO and observe the output activation. We start with a Type B residual block evaluated at input $\boldsymbol{x}^l = [\hat{\boldsymbol{x}}_+^l; \hat{\boldsymbol{x}}_-^l]$ at Layer $l$ and set $\alpha = 1$. The residual and skip branches at Layer $l$ are then

$$f^l(\boldsymbol{x}^l) = \mathbf{W}_2^l * \phi(\mathbf{W}_1^l * \boldsymbol{x}^l) = \mathbf{W}_2^l * \boldsymbol{x}^l = \begin{bmatrix} \mathbf{M}^l - (1/\alpha)\mathbb{I} & -\mathbf{M}^l \\ -\mathbf{M}^l & \mathbf{M}^l - (1/\alpha)\mathbb{I} \end{bmatrix} \begin{bmatrix} \hat{\boldsymbol{x}}_+^l \\ \hat{\boldsymbol{x}}_-^l \end{bmatrix}$$

$$= \begin{bmatrix} \mathbf{M}^l \hat{\boldsymbol{x}}_+^l - \mathbf{M}^l \hat{\boldsymbol{x}}_-^l - (1/\alpha)\hat{\boldsymbol{x}}_+^l \\ -\mathbf{M}^l \hat{\boldsymbol{x}}_+^l + \mathbf{M}^l \hat{\boldsymbol{x}}_-^l - (1/\alpha)\hat{\boldsymbol{x}}_-^l \end{bmatrix}.$$

Adding the skip branch to the residual branch gives

$$\alpha f^l(\boldsymbol{x}^l) + \boldsymbol{x}^l = \alpha \left[ \begin{array}{c} \mathrm{M}^l\hat{\boldsymbol{x}}^l_+ - \mathrm{M}^l\hat{\boldsymbol{x}}^l_- - (1/\alpha)\hat{\boldsymbol{x}}^l_+ \\ -\mathrm{M}^l\hat{\boldsymbol{x}}^l_+ + \mathrm{M}^l\hat{\boldsymbol{x}}^l_- - (1/\alpha)\hat{\boldsymbol{x}}^l_- \end{array} \right] + \left[ \begin{array}{c} \hat{\boldsymbol{x}}^l_+ \\ \hat{\boldsymbol{x}}^l_- \end{array} \right] = \alpha \left[ \begin{array}{c} \mathrm{M}^l\hat{\boldsymbol{x}}^l_+ - \mathrm{M}^l\hat{\boldsymbol{x}}^l_- \\ -\mathrm{M}^l\hat{\boldsymbol{x}}^l_+ + \mathrm{M}^l\hat{\boldsymbol{x}}^l_- \end{array} \right]$$

$$\phi(\alpha f^l(\boldsymbol{x}^l) + \boldsymbol{x}^l) = \alpha\phi\left( \left[ \begin{array}{c} \mathrm{M}^l\hat{\boldsymbol{x}}^l_+ - \mathrm{M}^l\hat{\boldsymbol{x}}^l_- \\ -\mathrm{M}^l\hat{\boldsymbol{x}}^l_+ + \mathrm{M}^l\hat{\boldsymbol{x}}^l_- \end{array} \right] \right) = \alpha \left[ \begin{array}{c} \hat{\boldsymbol{x}}^{l+1}_+ \\ \hat{\boldsymbol{x}}^{l+1}_- \end{array} \right].$$

The submatrices of the output of the residual block $\mathrm{U}^l_2\hat{\boldsymbol{x}}^l_+$ and $\mathrm{U}^l_2\hat{\boldsymbol{x}}^l_-$ preserve the norm of the input signal as long as we set $\alpha = 1$, since $\mathrm{U}^l_2$ is an orthogonal matrix fulfilling $\left\|\mathrm{U}^l_2\right\|^2 = 1$. If $\alpha$ takes a value other than 1 or -1, the orthogonal matrix $\mathrm{V}^l$ has to be scaled such that $\left\|\mathrm{U}^l_2\right\|^2 = 1/\alpha$. For all are experiments with type B residual blocks, however, we observe best results with $\alpha = 1$. Next we repeat the same computation for type C residual blocks. Note that the type C residual block can achieve an exactly orthogonal transform of the input with RISOTTO for any value of $\alpha$. The signal propagates through a residual block of type C initialized with RISOTTO and input $\boldsymbol{x}^l = \left[\hat{\boldsymbol{x}}^l_+; \hat{\boldsymbol{x}}^l_-\right]$ and weights

$$\mathrm{W}^l_1 = \left[ \begin{array}{cc} \mathrm{U}^l_1 & -\mathrm{U}^l_1 \\ -\mathrm{U}^l_1 & \mathrm{U}^l_1 \end{array} \right]; \mathrm{W}^l_2 = \left[ \begin{array}{cc} \mathrm{U}^l_2 & -\mathrm{U}^l_2 \\ -\mathrm{U}^l_2 & \mathrm{U}^l_2 \end{array} \right] \quad \mathrm{W}^l_{skip} = \left[ \begin{array}{cc} \mathrm{M}^l - \alpha\mathrm{U}^l_2\mathrm{U}^l_1 & -\mathrm{M}^l + \alpha\mathrm{U}^l_2\mathrm{U}^l_1 \\ -\mathrm{M}^l + \alpha\mathrm{U}^l_2\mathrm{U}^l_1 & \mathrm{M}^l - \alpha\mathrm{U}^l_2\mathrm{U}^l_1 \end{array} \right]$$

at Layer $l$ as

$$\alpha f^l(\boldsymbol{x}^l) + h^l(\boldsymbol{x}^l) = \alpha\mathrm{W}^l_2\phi(\mathrm{W}^l_1\boldsymbol{x}) + \mathrm{W}^l_{skip}\boldsymbol{x}$$

$$= \alpha \left[ \begin{array}{c} \mathrm{U}^l_2\phi\left(\mathrm{U}^l_1\hat{\boldsymbol{x}}^l_+ - \mathrm{U}^l_1\hat{\boldsymbol{x}}^l_+\right) - \mathrm{U}^l_2\phi\left(-\mathrm{U}^l_1\hat{\boldsymbol{x}}^l_+ + \mathrm{U}^l_1\hat{\boldsymbol{x}}^l_+\right) \\ -\mathrm{U}^l_2\phi\left(\mathrm{U}^l_1\hat{\boldsymbol{x}}^l_+ - \mathrm{U}^l_1\hat{\boldsymbol{x}}^l_+\right) + \mathrm{U}^l_2\phi\left(-\mathrm{U}^l_1\hat{\boldsymbol{x}}^l_+ + \mathrm{U}^l_1\hat{\boldsymbol{x}}^l_+\right) \end{array} \right]$$

$$+ \left[ \begin{array}{c} \phi\left((\mathrm{M}^l - \alpha\mathrm{U}^l_2\mathrm{U}^l_1)\hat{\boldsymbol{x}}^l_+ + (-\mathrm{M}^l + \alpha\mathrm{U}^l_2\mathrm{U}^l_1)\hat{\boldsymbol{x}}^l_-\right) \\ \phi\left(((-\mathrm{M}^l + \alpha\mathrm{U}^l_2\mathrm{U}^l_1))^l\hat{\boldsymbol{x}}^l_+ - (\mathrm{M}^l - \alpha\mathrm{U}^l_2\mathrm{U}^l_1)\hat{\boldsymbol{x}}^l_-\right) \end{array} \right]$$

$$= \left[ \begin{array}{c} \mathrm{M}^l\hat{\boldsymbol{x}}^l_+ - \mathrm{M}^l\hat{\boldsymbol{x}}^l_- \\ -\mathrm{M}^l\hat{\boldsymbol{x}}^l_+ + \mathrm{M}^l\hat{\boldsymbol{x}}^l_- \end{array} \right]$$

$$\boldsymbol{x}^{l+1} = \phi(\alpha f^l(\boldsymbol{x}^l) + h^l(\boldsymbol{x}^l)) = \left[ \begin{array}{c} \hat{\boldsymbol{x}}^{l+1}_+ \\ \hat{\boldsymbol{x}}^{l+1}_- \end{array} \right].$$

We conclude that for type C residual blocks the output is of looks-linear form and has the same norm as the input because of the orthogonal submatrices and the looks-linear structure.

We now use the above formulations to prove that RISOTTO preserves the squared signal norm and similarities between inputs for residual blocks, the key property that allows stable training.

**Theorem A.7** (**RISOTTO preserves signal norm and similarity**). *A residual block that is initialized with* RISOTTO *maps input activations* $\boldsymbol{x}^l$ *to output activations* $\boldsymbol{x}^{l+1}$ *so that the norm* $||\boldsymbol{x}^{l+1}||^2 = ||\boldsymbol{x}^l||^2$ *stays equal. The scalar product between activations corresponding to two inputs* $\boldsymbol{x}$ *and* $\tilde{\boldsymbol{x}}$ *are preserved in the sense that* $\langle\hat{\boldsymbol{x}}^{l+1}_+ - \hat{\boldsymbol{x}}^{l+1}_-, \tilde{\boldsymbol{x}}^{l+1}_+ - \tilde{\boldsymbol{x}}^{l+1}_-\rangle = \langle\hat{\boldsymbol{x}}^l_+ - \hat{\boldsymbol{x}}^l_-, \tilde{\boldsymbol{x}}^l_+ - \tilde{\boldsymbol{x}}^l_-\rangle$.

*Proof.* We first prove that the squared signal norms are preserved for both types of residual blocks followed by the same for similarity between inputs. Consider a type C residual block. The preactivation of the previous layer of looks linear form $\boldsymbol{z}^{l-1} = \left[\tilde{\boldsymbol{z}}^{l-1}; -\tilde{\boldsymbol{z}}^{l-1}\right]$. The preactivation of the current layer is then given as the signal passes through the residual block as

$$\boldsymbol{z}^l = \alpha f^l(\phi(\boldsymbol{z}^{l-1})) + h^l(\phi(\boldsymbol{z}^{l-1})) \tag{38}$$

$$= \alpha \left[ \begin{array}{c} \mathrm{U}^l_2\phi\left(\mathrm{U}^l_1\hat{\boldsymbol{x}}^l_+ - \mathrm{U}^l_1\hat{\boldsymbol{x}}^l_+\right) - \mathrm{U}^l_2\phi\left(-\mathrm{U}^l_1\hat{\boldsymbol{x}}^l_+ + \mathrm{U}^l_1\hat{\boldsymbol{x}}^l_+\right) \\ -\mathrm{U}^l_2\phi\left(\mathrm{U}^l_1\hat{\boldsymbol{x}}^l_+ - \mathrm{U}^l_1\hat{\boldsymbol{x}}^l_+\right) + \mathrm{U}^l_2\phi\left(-\mathrm{U}^l_1\hat{\boldsymbol{x}}^l_+ + \mathrm{U}^l_1\hat{\boldsymbol{x}}^l_+\right) \end{array} \right] \tag{39}$$

$$+ \left[ \begin{array}{c} \phi\left((\mathrm{M}^l - \alpha\mathrm{U}^l_2\mathrm{U}^l_1)\hat{\boldsymbol{x}}^l_+ + (-\mathrm{M}^l + \alpha\mathrm{U}^l_2\mathrm{U}^l_1)\hat{\boldsymbol{x}}^l_-\right) \\ \phi\left(((-\mathrm{M}^l + \alpha\mathrm{U}^l_2\mathrm{U}^l_1))^l\hat{\boldsymbol{x}}^l_+ - (\mathrm{M}^l - \alpha\mathrm{U}^l_2\mathrm{U}^l_1)\hat{\boldsymbol{x}}^l_-\right) \end{array} \right]$$

$$= \left[ \begin{array}{c} \mathrm{M}^l\hat{\boldsymbol{x}}^l_+ - \mathrm{M}^l\hat{\boldsymbol{x}}^l_- \\ -\mathrm{M}^l\hat{\boldsymbol{x}}^l_+ + \mathrm{M}^l\hat{\boldsymbol{x}}^l_- \end{array} \right] \tag{40}$$

$$= \left[ \begin{array}{c} \mathrm{M}^l\phi(\hat{\boldsymbol{z}}^{l-1}) - \mathrm{M}^l\phi(-\hat{\boldsymbol{z}}^{l-1}) \\ -\mathrm{M}^l\phi(\hat{\boldsymbol{z}}^{l-1}) + \mathrm{M}^l\phi(-\hat{\boldsymbol{z}}^{l-1}) \end{array} \right] \tag{41}$$

Then the squared norm is given by

$$\left\|\boldsymbol{z}^l\right\|^2 = \left\|\mathbf{M}^l\phi(\hat{\boldsymbol{z}}^{l-1}) - \mathbf{M}^l\phi(-\hat{\boldsymbol{z}}^{l-1})\right\|^2 + \left\|-\mathbf{M}^l\phi(\hat{\boldsymbol{z}}^{l-1}) + \mathbf{M}^l\phi(-\hat{\boldsymbol{z}}^{l-1})\right\|^2 \tag{42}$$

$$= 2\left\|\mathbf{M}^l\phi(\hat{\boldsymbol{z}}^{l-1})\right\|^2 + 2\left\|\mathbf{M}^l\phi(-\hat{\boldsymbol{z}}^{l-1})\right\|^2 \tag{43}$$

$$= 2\left(\left\|\phi(\hat{\boldsymbol{z}}^{l-1})\right\|^2 + \left\|\phi(-\hat{\boldsymbol{z}}^{l-1})\right\|^2\right) \tag{44}$$

$$= 2\left\|\hat{\boldsymbol{z}}^{l-1}\right\|^2 \tag{45}$$

$$= \left\|\boldsymbol{z}^{l-1}\right\|^2. \tag{46}$$

We have used the fact the $\mathbf{M}^l$ is an orthogonal matrix such that $\left\|\mathbf{M}^l x\right\| = \|x\|$ and the identity for ReLU activations by which $z = \phi(z) - \phi(-z)$. Since $\boldsymbol{x}^{l+1} = \phi(\boldsymbol{z}^l)$, we have

$$\boldsymbol{x}^{l+1} = \phi(\boldsymbol{z}^l) = \left[\begin{array}{c} \phi(\hat{\boldsymbol{z}}^l) \\ \phi(-\hat{\boldsymbol{z}}^l) \end{array}\right]. \tag{47}$$

Taking the squared norm of the output then gives

$$\left\|\boldsymbol{x}^{l+1}\right\|^2 = \left\|\phi(\hat{\boldsymbol{z}}^l)\right\|^2 + \left\|\phi(-\hat{\boldsymbol{z}}^l)\right\|^2 = \left\|\hat{\boldsymbol{z}}^l\right\|^2. \tag{48}$$

The second equality results from squaring the ReLU identity as the term $\langle\phi(\hat{\boldsymbol{z}}^l), \phi(-\hat{\boldsymbol{z}}^l)\rangle = 0$. Combining Eqns. 46 and 48, we obtain

$$\left\|\boldsymbol{x}^{l+1}\right\|^2 = \left\|\boldsymbol{x}^l\right\|^2. \tag{49}$$

$\square$

In fact, the norm preservation is a special case of the preservation of the scalar product, which we prove next.

Consider two independent inputs $\boldsymbol{x}$ and $\tilde{\boldsymbol{x}}$, their corresponding input activations at Layer $l$ are $\boldsymbol{x}^l$ and $\tilde{\boldsymbol{x}}^l$. We show that as a result of RISOTTO the correlation between the preactivations is preserved which in turn means that the similarity between activations of the looks linear form is preserved as $\langle\hat{\boldsymbol{x}}_+^{l+1} - \hat{\boldsymbol{x}}_-^{l+1}, \tilde{\hat{\boldsymbol{x}}}_+^{l+1} - \tilde{\hat{\boldsymbol{x}}}_-^{l+1}\rangle$

$$= \left[\phi(\hat{\boldsymbol{z}}^l) - \phi(-\hat{\boldsymbol{z}}^l)\right]^T \left[\phi(\tilde{\hat{\boldsymbol{z}}}^l) - \phi(-\tilde{\hat{\boldsymbol{z}}}^l)\right] \tag{50}$$

$$= \left[\phi\left(\mathbf{M}^l\phi(\hat{\boldsymbol{z}}^{l-1}) - \mathbf{M}^l\phi(-\hat{\boldsymbol{z}}^{l-1})\right) - \phi\left(-\mathbf{M}^l\phi(\hat{\boldsymbol{z}}^{l-1}) + \mathbf{M}^l\phi(-\hat{\boldsymbol{z}}^{l-1})\right)\right]^T$$
$$\times \left[\phi\left(\mathbf{M}^l\phi(\tilde{\hat{\boldsymbol{z}}}^{l-1}) - \mathbf{M}^l\phi(-\tilde{\hat{\boldsymbol{z}}}^{l-1})\right) - \phi\left(-\mathbf{M}^l\phi(\tilde{\hat{\boldsymbol{z}}}^{l-1}) + \mathbf{M}^l\phi(-\tilde{\hat{\boldsymbol{z}}}^{l-1})\right)\right] \tag{51}$$

$$= \left[\left(\phi(\mathbf{M}^l\phi(\hat{\boldsymbol{z}}^{l-1})) - \phi(-\mathbf{M}^l\phi(\hat{\boldsymbol{z}}^{l-1}))\right) - \left(\phi(\mathbf{M}^l\phi(-\hat{\boldsymbol{z}}^{l-1})) - \phi(-\mathbf{M}^l\phi(-\hat{\boldsymbol{z}}^{l-1}))\right)\right]^T$$
$$\times \left[\left(\phi(\mathbf{M}^l\phi(\tilde{\hat{\boldsymbol{z}}}^{l-1})) - \phi(-\mathbf{M}^l\phi(\tilde{\hat{\boldsymbol{z}}}^{l-1}))\right) - \left(\phi(\mathbf{M}^l\phi(-\tilde{\hat{\boldsymbol{z}}}^{l-1})) - \phi(-\mathbf{M}^l\phi(-\tilde{\hat{\boldsymbol{z}}}^{l-1}))\right)\right] \tag{52}$$

$$= \left[\mathbf{M}^l\left(\phi(\hat{\boldsymbol{z}}^{l-1}) - \phi(-\hat{\boldsymbol{z}}^{l-1})\right)\right]^T \left[\mathbf{M}^l\left(\phi(\tilde{\hat{\boldsymbol{z}}}^{l-1}) - \phi(-\tilde{\hat{\boldsymbol{z}}}^{l-1})\right)\right] \tag{53}$$

$$= \left[\phi(\hat{\boldsymbol{z}}^{l-1}) - \phi(-\hat{\boldsymbol{z}}^{l-1})\right]^T \left[\phi(\tilde{\hat{\boldsymbol{z}}}^{l-1}) - \phi(-\tilde{\hat{\boldsymbol{z}}}^{l-1})\right] \tag{54}$$

$$= \langle\hat{\boldsymbol{x}}_+^l - \hat{\boldsymbol{x}}_-^l, \tilde{\hat{\boldsymbol{x}}}_+^l - \tilde{\hat{\boldsymbol{x}}}_-^l\rangle \tag{55}$$

The above derivations follow from the looks -linear structure of the weights and the input as well as the orthogonality of matrix $\mathbf{M}^l$. The same proof strategy for both norm preservation and similarity can be followed for type B residual blocks using the signal propagation A.3. This concludes the proof.

## A.4 EXPERIMENTAL SETUP AND DETAILS

In all our experiments we use Stochastic Gradient Descent (SGD) with momentum 0.9 and weight 0.0005. We use 4 NVIDIA A100 GPUs to train all our models. All experiments are repeated for

3 runs and we report the mean and $0.95$ confidence intervals. In experiments with ResNet101, we used a learning rate of $0.005$ for all initialization schemes including ours.

We performed the following hyperparameter tuning for each initialization scheme + model + dataset that we compare and report. For every experiment we use the best performing learning rate between $0.1, 0.01, 0.05, 0.001$ with either cosine annealing or step scheduling (whichever performs better), while keeping the batch size fixed to 256. To do so, we randomly split the training set into a training (90%) and a validation (10%) set and used the LR that performed best on the validation set to evaluate the trained model on the test set. We report the accuracies obtained on the test set.

**Placing a single BN layer**   In order to identify the best position to place a single BN layer in a ResNet, we experiments with 3 different positions. $(i)$ First layer: The BN was placed right after the first convolution layer before the residual blocks. $(ii)$ BN in the middle: The BN layer was placed after half of the residual blocks in the network. $(iii)$ BN after last res block: In this case BN was placed before the pooling operation right after the last residual block.

**Correlation comparison in Figure 2b**   In order to compare the correlation between inputs for different initialization schemes we use a vanilla Residual Network with five residual blocks, each consisting of the same number of channels (32) and a kernel size of $(3, 3)$ followed by n average pooling and a linear layer. The figure shows the correlation between two random samples of CIFAR10 averaged over 50 runs.

**Tiny Imagenet**   Note that we use the validation set provided by the creators of Tiny Imagenet (Le & Yang, 2015) as a test set to measure the generalization performance of our trained models.

| | | without BN | | | with BN | | |
|---|---|---|---|---|---|---|---|
| Type | Param | RISOTTO (ours) | Fixup | SkipInit | RISOTTO (ours) | He Uniform | He Normal |
| | LR | 0.1 | 0.01 | 0.01 | 0.1 | 0.1 | 0.1 |
| C | BS | 256 | 256 | 256 | 256 | 256 | 256 |
| | Schedule | cosine | cosine | cosine | cosine | cosine | cosine |
| | Epochs | 150 | 150 | 150 | 150 | 150 | 150 |
| | LR | 0.1 | 0.05 | 0.05 | 0.1 | 0.1 | 0.1 |
| B | BS | 256 | 256 | 256 | 256 | 256 | 256 |
| | Schedule | cosine | cosine | cosine | cosine | cosine | cosine |
| | Epochs | 150 | 150 | 150 | 150 | 150 | 150 |

Table 3: Implementation details for ResNet18 on CIFAR10

| | | without BN | | | with BN | | |
|---|---|---|---|---|---|---|---|
| Type | Param | RISOTTO (ours) | Fixup | SkipInit | RISOTTO (ours) | He Uniform | He Normal |
| | LR | 0.01 | 0.01 | $0.001^*$ | 0.1 | 0.1 | 0.1 |
| C | BS | 256 | 256 | 256 | 256 | 256 | 256 |
| | Schedule | cosine | cosine | cosine | cosine | cosine | cosine |
| | Epochs | 150 | 150 | 150 | 150 | 150 | 150 |
| | LR | 0.01 | 0.01 | 0.01 | 0.1 | 0.1 | 0.1 |
| B | BS | 256 | 256 | 256 | 256 | 256 | 256 |
| | Schedule | cosine | cosine | cosine | cosine | cosine | cosine |
| | Epochs | 150 | 150 | 150 | 150 | 150 | 150 |

Table 4: Implementation details for ResNet50 on CIFAR100. $*$ denotes that SkipInit failed to train even at a very low learning rate for multiple runs as reported in Table 1 in the main paper.

| | | without BN | | | with BN | | |
|---|---|---|---|---|---|---|---|
| Type | Param | RISOTTO (ours) | Fixup | SkipInit | RISOTTO (ours) | He Uniform | He Normal |
| C | LR | 0.01 | 0.01 | 0.001 | 0.01 | 0.01 | 0.01 |
| | BS | 256 | 256 | 256 | 256 | 256 | 256 |
| | Schedule | cosine | cosine | cosine | step $(30, 0.1)$ | step $(30, 0.1)$ | step $(30, 0.1)$ |
| | Epochs | 150 | 150 | 150 | 150 | 150 | 150 |
| B | LR | 0.01 | 0.01 | 0.01 | 0.01 | 0.01 | 0.01 |
| | BS | 256 | 256 | 256 | 256 | 256 | 256 |
| | Schedule | cosine | cosine | cosine | cosine | cosine | cosine |
| | Epochs | 150 | 150 | 150 | 150 | 150 | 150 |

Table 5: Implementation details for ResNet50 on Tiny Imagenet. Arguments for step denote that learning rate was reduced by a factor of 0.1 every 30 epochs.

## A.5 ADDITIONAL EXPERIMENTS FOR BN LAYER PLACEMENT ON CIFAR100

Considering the importance of BN on performance and how other methods struggle to compete with it, we empirically explore if reducing the number of BN layers in a network would still render similar benefits at reduced computational and memory costs. We observe that in such a case, the position of the single BN layer in the network plays a crucial role. Figure 5 shows that RISOTTO enables training in all cases while other initializations fail if the single BN layer is not placed after the last convolutional layer to normalize all features. First we need to determine the best position to place the BN layer in the network. We report experiments in Figure 5 that placing the BN after the last convolutional layer, before the last linear layer achieves best results. In Figure 7 we show results for Tiny Imagenet with a single BN after the last residual layer where too RISOTTO outperforms the standard. BN after the last residual layer controls the norms of the logits and potentially stablizes the gradients leading to better performance. Conversely, BN right after the first layer does not enable larger learning rates or better generalization. In Figure 7, we show that even after optimal placement of the single BN layer, RISOTTO leads to the overall best results on Tiny ImageNet at reduced computational costs.

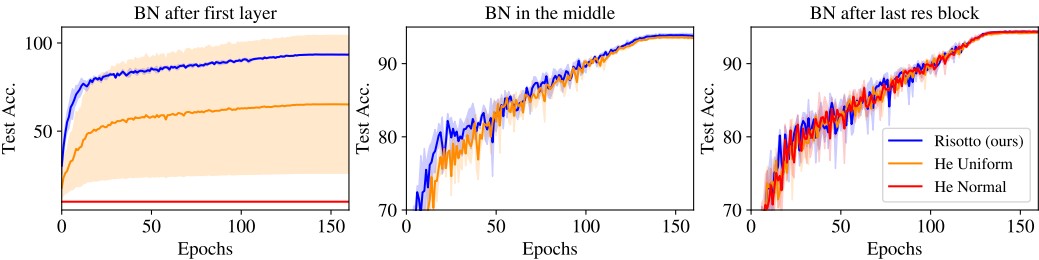

Figure 5: Comparing different positions of placing a single BN layer on a ResNet 18 (C) for CI-FAR10. In each of the cases, RISOTTO allows stable training and converges to competetive accuracies, while standard methods fail in some cases.

## A.6 TRAINING VERY DEEP NETWORKS WITH RISOTTO

We train a Residual Network with 300 layers (and no BN) initialized with Risotto and show that with Risotto, such a deep network is able to train faster compared to Fixup (see Figure 8).

## A.7 COMPARISON WITH NORMALIZATION FREE RESNETS

**Comparison with Normalization Free ResNets** NF ResNets (Brock et al., 2021a) that use scaled weight standardization and have been shown to compete with BN. We find that RISOTTO is able to match or outperform NF ResNets with He initialization He et al. (2016) as shown in Fig. 9. While

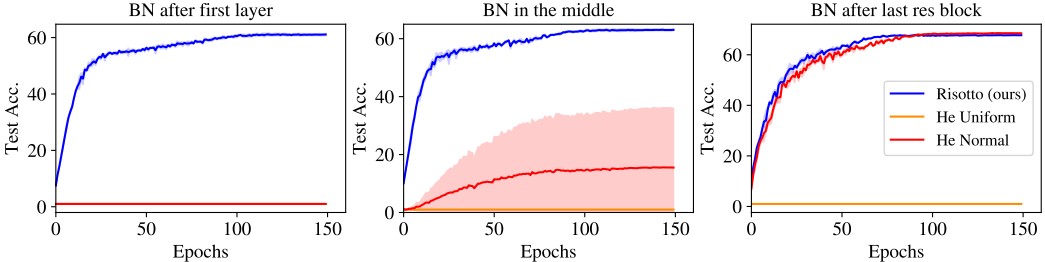

Figure 6: Comparing different positions of placing a single BN layer on CIFAR100 for a ResNet50 (C). In each case, RISOTTO allows stable training and converges to competitive test accuracies, while standard methods fail in some cases. Standard methods are more unstable in this case compared to CIFAR10 and only He Normal is competetive when BN is placed in the last layer.

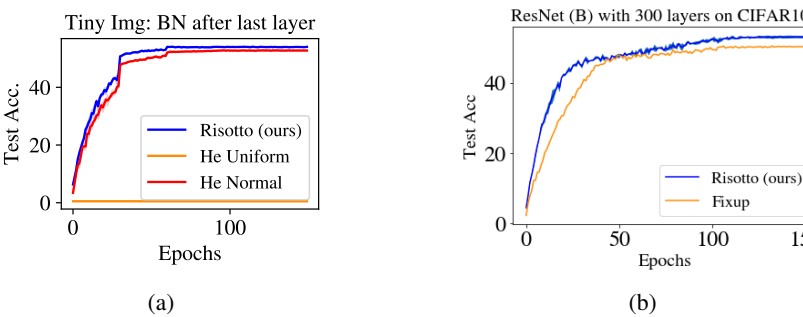

(a)                                                                (b)

Figure 7: ($a$) Using only one layer of BN after the last residual block in a ResNet 50 (C) on Tiny ImageNet. RISOTTO still achieves accuracy competitive to BN in every layer and performs better than Normal He initialization, while He Uniform initialization fails completely. ($b$) A Residual Network (type B) with 300 layers, which is a deeper version of a ResNet101 with Bottleneck blocks arranged as $[3 * 3, 3 * 4, 3 * 23, 3 * 3]$, is trained with Risotto and Fixup (both with a LR $= 0.01$) and cosine annealing for 150 epochs on CIFAR100. The rest of the hyperparameters are as in Table 4. We observe that Risotto enable faster training and better generalization.

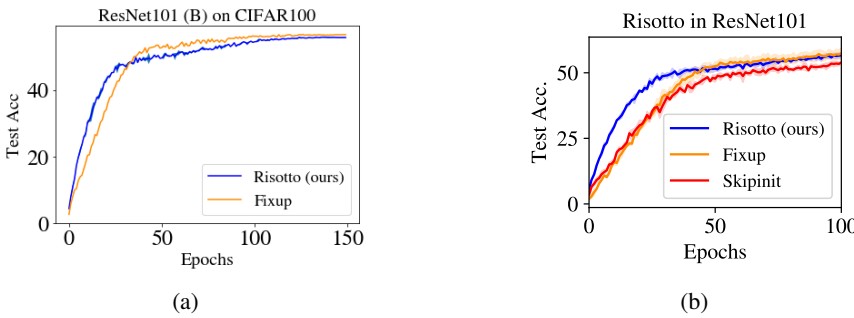

(a)                                                                (b)

Figure 8: (a) Similar to Figure 8b, we compare Risotto without BN against Fixup on a ResNet101. Here we use an LR $= 0.01$ for both methods. (b) ResNet101 (C) on CIFAR100. Training with RISOTTO achieves faster a good performance.

NF ResNets usually require careful hyperparameter tuning for gradient clipping, we observe that they train well with vanilla SGD on smaller datasets.

## A.8    RESULTS ON IMAGENET

We also report experiments on the large scale ImageNet dataset. Due to limited computational resources, we are only able to report a single run on ResNet50 (B), but our results are in line with our

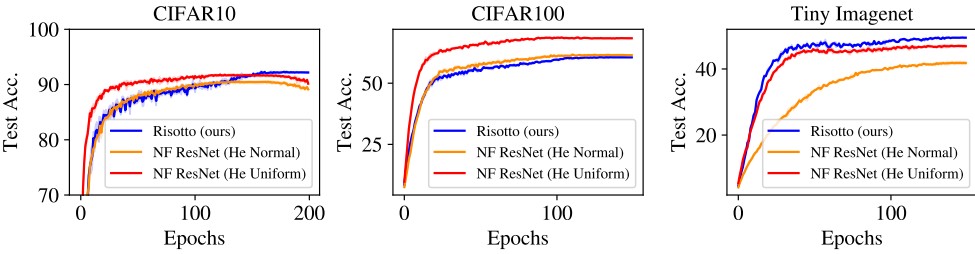

Figure 9: Comparing NF ResNets to RISOTTO on all three datasets with a ResNet (C) (18 for CIFAR10 and 50 for the others). While RISOTTO performs similar to He Normal on CIFAR100, it is able to outperform both He Uniform and He Normal on Tiny ImageNet and CIFAR10.

findings on Tiny ImageNet. When we use our initialization for ImageNet in which we add a small amount of noise $\epsilon \Sigma$ to our RISOTTO parameters, where $\epsilon = 10^{-4}$ and $\Sigma$ follows the He Normal distribution (He et al., 2015) for the corresponding Kaiming Normal initialization. Adding this random noise enables symmetry breaking (Blumenfeld et al., 2020) and supports training according to our empirical observations. Results are reported in Figure 10.

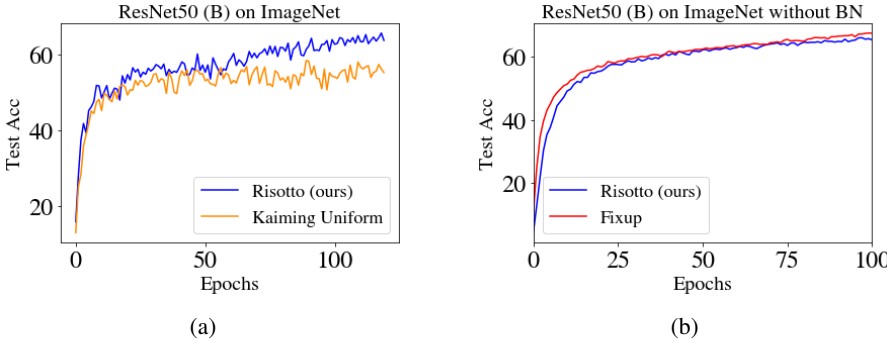

| | |
|---|---|
| (a) | (b) |

Figure 10: (a) We train a ResNet50 (B) with BN and compare Risotto (ours) with the standard Kaiming Normal initialization scheme. We train both the models with LR = 0.01 and cosine annealing. We observe that with our initialization scheme we are able to train faster on ImageNet and even achieve better accuracy than Kaiming Normal. (b) Similarly, we compare Risotto (ours) with Fixup on a ResNet50 (B) without BN. Both are trained with LR = 0.01. Risotto tracks Fixup closely.

**Remark**: We would like to note that because our results on ImageNet are from a single run, they are inherently susceptible to variance between runs. Nonetheless, we have comprehensively shown the effectiveness of RISOTTO on multiple datasets.

## A.9    RESULTS ON PREACTIVATION RESNETS

In addition to the ResNet models we have considered so far in the paper, there is another set of models in Residual Networks that may be used for certain problems. Preactivation ResNets (He et al., 2015) apply the ReLU nonlinearlity before the weight within a residual block. Our initialization scheme, Risotto, can be easily transferred to Preactivation ResNets in order to achieve dynamical isometry. We report experiments on CIFAR10 for Preactivation ResNets in Figure 11.

## A.10    ADDITIONAL EXPERIMENTS WITH RESNET (B) FOR RISOTTO

We provide additional experiments showing the effectiveness of Risotto for ResNet(B) architectures in Figure 12.

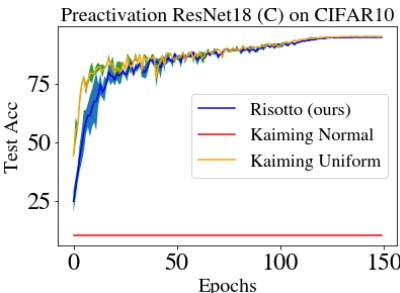 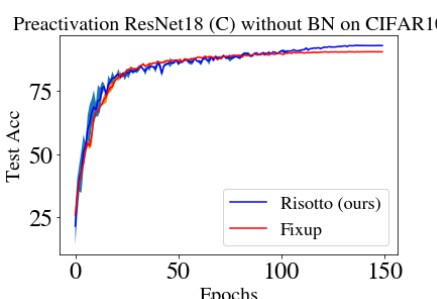

Figure 11: We train a Preactivation ResNet18 (C) with BN initialized with Risotto (ours) and compare with the standard Kaiming Normal and Kaiming Uniform initialization schemes. We train both the models with LR = 0.1 and cosine annealing. Risotto is able to enable dynamical isometry and achieve competetive generalization. Similarly we also report results withour BN comparing Risotto with Fixup, where Risotto is able to outperform Fixup.

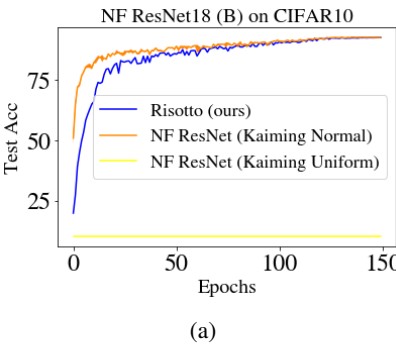 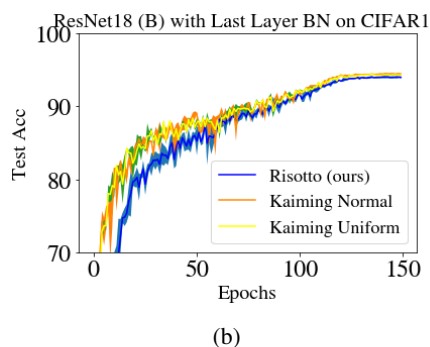

(a)                                                      (b)

Figure 12: (a) NF-ResNet18 (B) compared with ResNet 18 (B) initialized with Risotto (ours) and without any Batch Normalization. (b) Using a single layer of BN after the last residual block in a ResNet18 (B) on CIFAR10.

## A.11   TRAINING A 2000 LAYER RESIDUAL NETWORK

To show that Risotto enables training very deep networks due to initial dynamical isometry, we train a very deep residual network with Risotto. We use a Residual Network with Bottleneck residual blocks (Type C) arranged in the form $[20*3, 20*4, 20*23, 20*3]$ with channels $[64, 128, 256, 512]$ respectively. We use a learning rate of $5e-4$ and add noise during initialization similar to A.9. We see that initializing with Risotto enable training even a very deep network. While Risotto enables training such a deep network, we would like to remark that training such a deep network to convergence and a competetive performance would require additional methods like gradient clipping to ensure gradients do not vanish or explode during training.

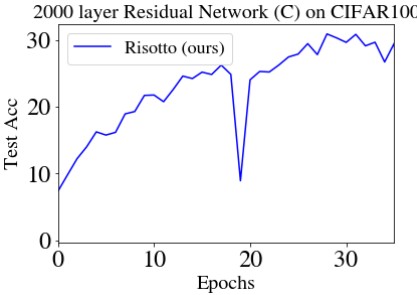

Figure 13: Risotto enables training a 2000 layer network on CIFAR100.

## A.12 RISOTTO WITH NOISE: N-RISOTTO

We also report results of RISOTTO initialized with additional noise. The model is initialized with RISOTTO in which we add a small amount of noise $\epsilon\Sigma$ to our RISOTTO parameters, where $\epsilon = 10^{-4}$ and $\Sigma$ follows the He Normal distribution (He et al., 2015) For the CIFAR10/100 experiments with N-RISOTTO we use a learning rate of $0.05$ with cosine annealing, except for CIFAR100 with ResNet50 (B) where we are able to use a higher learning rate of $0.1$. For Tiny ImageNet we use a learning rate of $0.01$ with cosine annealing.

| Dataset | ResNet | RISOTTO + noise (ours) with BN | RISOTTO + noise (ours) without BN |
|---|---|---|---|
| CIFAR10 | 18 (C) | 94.88 | 91.88 |
| | 18 (B) | 95.06 | 91.57 |
| CIFAR100 | 50 (C) | 73.22 | 60.29 |
| | 50 (B) | 77.74 | 64.46 |
| Tiny ImageNet | 50 (C) | 58.82 | 49.66 |
| | 50 (B) | 59.26 | 47.09 |

Table 6: **N-RISOTTO** We initialize networks with RISOTTO and add a small amount of noise to enable 'breaking the symmetry' (Blumenfeld et al., 2020).

