# OpenReview forum: "Dynamical Isometry for Residual Networks"
_ICLR.cc/2023/Conference — Submitted to ICLR 2023_

### Official Review · Reviewer_VqzH · 2022-10-24

**Confidence:** 5
**Correctness:** 3
**Technical Novelty And Significance:** 2
**Empirical Novelty And Significance:** 3
**Recommendation:** 6

**Clarity, Quality, Novelty And Reproducibility:**

There are several aspects that the paper can improve in writing:
- It is worth carefully rethinking whether the experiments of adding BN are related to the central story and how to make the point.
- Sections 2.1 and 2.2 can be made much more concise. I got distracted when I read this section. It would be better if the authors can first clearly state what they propose to change, and then discuss the background and motivation. It would also be helpful to add figures or an algorithm box to emphasize your modifications.

This paper is novel since the proposed method has not been considered before. It also provides a way of dealing with ReLU non-linearities.

It provides code. I checked the relevant implementation. I think the proposed method is simple and also easy to reproduce.

**Details Of Ethics Concerns:**

Not Applicable.

**Strength And Weaknesses:**

Strength:
1) The proposed method is simple and effective. It makes ResNet without normalization trainable just by changing the initialization of convolution. Compared to fixup, it does not initialize the output of the residual branch to be 0 (as in Fixup). This is surprising to me.
2) It takes explicit consideration of ReLU activation and achieves exact dynamical isometry for residual blocks. This is different from several previous works and can be inspiring for future research.

Weakness:
1) The central theme of this paper is unclear to me. From the introduction, I think the authors want to describe the disadvantage of batch normalization and propose a way to remove batch normalization, which is shown in Table 1. However, the authors also use several paragraphs and Table 2 to show RISOTTO is compatible with batch normalization. It makes me confused because batch normalization can reduce the sensitivity of the initialization method. It is not clear why showing the compatibility between RISOTTO and BN can support the paper’s claim.
2) In Table 1 and Table 2, it seems the proposed method is not better than the previous method (fixup) without batch normalization.
3) This paper mentions “separability of inputs” several times but I didn’t understand what it means. The authors may want to consider adding the reference to that part. Is it equivalent to maintaining the inner product of input?
4) Missing reference: This paper claims previous work only proposes dynamical isometry in a vanilla network (without residual structures). In [1], the paper proposes a way to achieve “approximate dynamic isometry” both at initialization and training time. There are differences between [1] and the proposed work such as [1] initializing the output of residual blocks to be 0 which is opposite to this work, but it’s worth comparing and discussing.
5) The proposed method only shows the trainability of a network with 50 layers on CIFAR. In contrast, previous methods usually train 100 layers (fixup and [1]) on ImageNet or 10000 layers on CIFAR. Whether it can be scaled to deeper networks remains unknown. The proposed method is also only validated on CIFAR while many previous methods can train deep resnet on ImageNet without a normalization layer (fixup and [1]).
[1] Qi et al. Deep Isometric Learning for Visual Recognition. ICML 2020.

**Summary Of The Paper:**

This paper proposes a new initialization method for ConvNets with residual blocks and ReLU activation, called RISOTTO. It enables training deep networks with residual structures without a normalization layer. This is achieved by maintaining a dynamical isometry at the initialization time, which is implemented as initializing the weight matrices as a block-wise looks-linear structure. The experiments on CIFAR and Tiny Imagenet can support the claim of this paper.

**Summary Of The Review:**

In summary, this paper is interesting as it shows by designing a simple initialization method, deep networks with residual structures can be trained. On the other hand, the content of this paper can be improved a lot. I recommend acceptance for this paper but I expect to see an improved version of this paper in the discussion phase.

---

> ### Author Response · Authors · 2022-11-13
> **Reply to Reviewer VqzH**
>
> We thank Reviewer VqzH for the in-depth review and constructive feedback.
>
> **Central theme**
>
> Our main objective was to achieve dynamical isometry at initialization for ResNets while giving similar weight to skip and residual connections and promoting feature diversity. Naturally, this combination of multiple objectives also has multiple advantages that go beyond just the classic motivation for dynamical isometry, which is technically also achieved by Fixup, SkipInit, ReZero, R-ISONet, etc., but at the expense of low feature diversity and scaled down residual connections. We believe that we do not have to focus on one single advantage for a round story, as our contribution is very clearly focused on our initialization scheme Risotto. We have adapted our introduction and discussion and added Figure 1 for an overview to clarify how the following advantages of Risotto are related:
>
> 1. The standard motivation for initial dynamical isometry is that it enables training of very deep networks without BN and often speeds up training. This point is particularly important for labs that want to train large scale models on big datasets. The mentioned papers that either train very deep networks (consisting of 10000 layers) or discuss mostly large scale experiments on ImageNet were authored by researchers from big vision labs that primarily have this application scenario in mind. Their focus on large scale experiments is also reasonable because the benefits of their proposed orthogonal initialization schemes become apparent in this setting, while they make less of a difference in smaller scale problems. We also provide different types of evidence that Risotto could create impact in this scenario:
> - Our theory and the cited literature suggest that Risotto enables training of very deep networks without BN because it induces perfect initial dynamical isometry, while initialization schemes with independent weight entries cannot achieve this without BN.
> - We show that training 2000 layer neural networks with Risotto without BN is feasible (and outperforms Fixup) on CIFAR100 (see Figure 13 Appendix A.11).
> - We have added the requested experiments on ImageNet that show that Risotto enables training of ResNets without BN on large scale data. (These are in line with our findings on Tiny ImageNet.)
>
> However, we should not forget that training large scale models on big data is not the only relevant application scenario of deep learning. For instance, many problems in the biomedical domain are of smaller scale but equally important to lead progress in AI. Furthermore, proposals that only innovate the initialization scheme without BN were so far not able to compete with BN when it actually can be used.
>
> 2. Special about our results is that we can also show improvements in smaller scale scenarios and even in combination with BN, which is usually less sensitive to the initialization scheme. We hypothesize that the reason for that is that we give higher weight to the residual connections. Note that even standard He initialization approaches in combination with BN tend to scale down the residual connections according to [De and Smith](https://arxiv.org/abs/2002.10444). So do all other ResNet initialization schemes with the dynamical isometry property. In contrast, Risotto balances residual and skip connections. The advantages of this approach (and the effect of alternative scalings) are visualized in Figure 1c). Our experiments with BN also highlight the benefits of this aspect of Risotto.
>
> 3. (Now in the appendix:) Considering the fact that BN usually leads to excellent training and generalization results, which is even improved by Risotto, it would be great if we could use it in more application scenarios (coming back to our first objective). Reducing the number of BN layers and being flexible in placing them at different positions could enable us to use BN in lower dimensional layers (even if the input is so high dimensional that we cannot use large batch sizes in early layers because of memory constraints). As we show, Risotto gives us full flexibility in the location of BN layer placement. The reason is that it enables anyways training very deep neural networks (as discussed in 1)) without BN layers. Furthermore, it works well in combination with BN (as discussed in 2)), which makes it the right candidate to experiment with different BN layer placements.
>
> In summary, Risotto is flexible in its combination with BN and can also be applied in scenarios that prevent the use of BN, as proven by the fact that it has the dynamical isometry property. It is therefore based on a sound theoretical foundation.

---

> > ### Author Response · Authors · 2022-11-13
> > **Reply to Reviewer VqzH (continued)**
> >
> > **Risotto leads almost always to the best overall result**
> >
> > Note that Risotto with ResNet C always leads to the overall best result in Table 1. In Table 2, the overall best average result is always achieved on CIFAR100 and Tiny ImageNet by Risotto. The only exception is the small-scale ResNet18 on CIFAR10 with BN, but even in this case Risotto is almost on par with the best result.
> >
> > **Separability of inputs**
> >
> > The reviewer is correct with the assumption that we refer to the dot product between two input activations when we talk about the separability of inputs, as shown in Insight 2.5 and 2.11. We clarify this now earlier in our updated draft. We analyze separability of inputs to show that standard initialization schemes with independent parameters map different inputs to similar features in a deep network, whereas an orthogonal scheme like Risotto maintains a constant dot product between input activations for any depth.
> >
> > **Additional reference**
> > We thank Reviewer VqzH for pointing out this very interesting work on [ISONet](https://arxiv.org/abs/2006.16992) that we have added to the literature discussion and outlook. We have decided to not add R-ISONet as a full baseline in our experiments because it changes the activation function to shifted ReLUs with additional trainable bias parameters. Otherwise, we have restricted our experiments to initialization schemes for ReLUs and feel that a comparison with R-ISONet would dilute our story. However, to satisfy the curiosity of Reviewer VqzH and our own, we have initialized ResNet50 B with the proposed R-ISONet initialization and trained the network on CIFAR100 (without BN). It achieves a test accuracy of 62.83, while Risotto leads to 64.52.
> > We still see great potential for using the proposed regularization towards orthogonality with Risotto and, vice versa, transferring our idea to balance skip and residual connections to initialize ResNets with the proposed shifted ReLU activation functions, as we discuss as an outlook in our updated manuscript.
> >
> > **Large scale experiments**
> > We have added a couple of experiments to the appendix (that were possible with our computing infrastructure).
> > We have complemented our experiments on ResNet101 with much deeper ResNets consisting of 300 layers (see Figure 4 (center)). Our initialization Risotto trains faster than Fixup and achieves a higher test accuracy on CIFAR100. We also show that a 2000 layer Residual Network is trainable with Risotto on CIFAR100 (see Figure 13, in Appendix A.11).
> > We have added ImageNet experiments (see Figure 10 in Appendix A.8). After 120 epochs, a single run of ResNet 50 (B) with BN achieves 65.66% test accuracy for Risotto and 58.41 test accuracy for a baseline He initialization with BN. Without BN, ResNet 50 (B) achieves 67.05% test accuracy with Risotto, while Fixup achieves 67.05% test accuracy.
> >
> > **Additional figures that highlight the main features/contributions of Risotto**
> > We have updated our manuscript with an additional overview figure (see Figure 1) that highlights the three main features of Risotto (dynamical isometry, feature diversity, balanced skip and residual connections). We have also added Figure 3 to explain the Risotto approach visually. We thank Reviewer VqzH for the great suggestion. To make space, we have therefore moved the experiments on BN layer placement to the appendix.

---

> > > ### Comment · Reviewer_VqzH · 2022-11-20
> > > **Response**
> > >
> > > I thank the authors for their detailed response to my review and the additional expeirments. In summary, I still lean on acceptance but there are still many places can be improved (thus my recommentation is still marginally above the acceptance threshold).
> > >
> > > Regarding the story: it becomes more clear after the author explains their motivation. I think the similar development should be added into the main paper. I also agree some smaller scale experiments are important for sure, but if the authors want to claim your method is beneficial, you need to show experiments to support such a claim.
> > >
> > > Regarding large scale experiments: the authors successfully show the trainability of a deeper network or on a larger dataset. However, the performance is not very reasonable: In Figure 10 (a), the accuracy of a resnet 50 is only ~60, which is even significantly lower than the original resnet. This is why I cannot further increase my score.

---

> > > > ### Author Response · Authors · 2022-11-21
> > > > **We have not optimized the data augmentation for ImageNet**
> > > >
> > > > We appreciate the quick response by Reviewer VqzH and will try to address the fair criticism regarding the ResNet accuracy on ImageNet.
> > > >
> > > > The reasoning behind our conducted experiments was that we wanted to maintain consistency. We did not want to change the experimental set-up for ImageNet in comparison with our experiments on CIFAR and Tiny ImageNet. Note that the standard reported ImageNet accuracy crucially depends on the used data augmentation strategy (like mixup, etc.)
> > > > For that reason, we also computed baseline results on ImageNet with the same experimental set-up (but with optimized learning rate, etc., as reported) that demonstrate that Risotto is able to achieve improvements. We feel that tuning also data augmentation strategies now would shift the focus away from the effect of initialization.
> > > >
> > > > We would also like to emphasize that our statements regarding the trainability of very deep networks (which is more relevant on large scale complex data) is backed up by theory and experiments with a 2000 layer ResNet. We never claim or claimed that dynamical isometry is necessarily of benefit for large scale data. It induces benefits for training very deep ResNets. Our story does not crucially depend on ImageNet.

---

> > > > ### Author Response · Authors · 2022-11-26
> > > > **Tuning Imagenet Experiments**
> > > >
> > > > In order to show that Risotto is indeed able to achieve competitive performance and can be easily integrated into existing experimental setups, we train Risotto in an environment  that was tuned for our baselines (and not Risotto itself). With BN, we achieve an accuracy of 75.7% in 100 epochs on ImageNet with a ResNet50 (B). Without BN, we achieve an accuracy of 71.4% after 90 epochs which is at par with Fixup.

---

### Official Review · Reviewer_GPrm · 2022-10-25

**Confidence:** 4
**Correctness:** 2
**Technical Novelty And Significance:** 2
**Empirical Novelty And Significance:** 1
**Recommendation:** 3

**Clarity, Quality, Novelty And Reproducibility:**

Clarity: Medium.

Quality: limited.

Novelty: Medium.

Reproducibility: Provided codes.

**Strength And Weaknesses:**

Pros:

- Writing is comparably easy to follow.

Cons:

- The main difference between the proposed RISOTTO and other initialization (e.g., fixup) is that RISOTTO has analyzed the weight in residual connection (termed as C type in this paper). However, This C type is not the general case in residual nets. In a residual net, there can be a pooling layer for downsampling. And even in a case using C type, the weight is only used for downsampling, thereby not as general as no weight in residual connection. Moreover, Transformer is also a type of residual network, they do not have such a C type. Therefore, I feel that the motivation is not sufficient.

- On experiments:
  - The superiority of RISOTTO is not significant. For example, in Table 1 and Table 2, all the baselines (i.e., Fixup, SkipInit, He Normal, and He Uniform) can derive better results in some settings. Especially, Fixup achieves all better results on B type, the more general condition.
  - The ResNets are trained for 150 epochs may not be enough. It is common to train 200 epochs at least.
  - In Table 2, the results of two He initialization on Tiny ImageNet are incredible. Specifically, on B type, He Normal derives 52.21, and He Uniform derives 55.05. There is around a 3% distance. It is hard to believe the results as these two initialization are both in accordance with He initialization, and the only difference is the sampling strategy.
  - A learning rate of 0.1 is a common setting. Why did this paper set 0.01 for training ResNet50 on Tiny Imagenet, while 0.1 for cifar10/100?
  - Figure 2, Figure 3, and Figure 4 are all based on C-type nets. What are the results of B type as it is also an important architecture?
  - In Figure 3, Fixup seems to derive better results.
  - It is better to add Transformer as a backbone.
  - It is better to add experiments on large-scale datasets, like ImageNet.


Overall, regarding the above points, I vote for "rejection".

**Summary Of The Paper:**

The paper proposed RISOTTO (Residual dynamical isometry by initial orthogonality) to initialize residual networks. While RISOTTO achieves some success in several datasets. I feel that the experiments are inadequate, and the superiority of RISOTTO is not significant.

**Summary Of The Review:**

The paper proposed an isometry dynamic based initialization. However, I still have some concerns as mentioned above. Therefore, I tend to give "rejection".

---

> ### Author Response · Authors · 2022-11-13
> **Reply to Reviewer GPrm**
>
> We thank Reviewer GPrm for their feedback on our work.
>
> **Architecture**
> Our initialization scheme Risotto is not limited to Type C residual blocks. For instance, it covers Type B blocks as well and could also handle downsampling similar to how we treat Type B blocks.
> Risotto could also improve the initialization of transformers by following a similar approach as ReZero by [Bachlechner et al.](https://arxiv.org/abs/2003.04887)  However, this is out of scope of our work, since our whole theory and exposition is focused on ResNets, which are still quite popular deep learning architectures.
>
> **Experiments**
>
> 1. Superiority of Risotto: One variant of Risotto wins always (except for a single small-scale case for ResNet18 with BN on CIFAR10). When we increase the depth and complexity of the dataset, the benefits of Risotto become more clear. It is superior in all cases on Tiny Imagenet as seen in Tables 1 and 2. Also on ImageNet, we find that Risotto outperforms He initialization with BN, and Fixup without BN.
> 2. We experimented with training models for 250 epochs and observed convergence at ~ 130 epochs. Hence, we stick to 150 epochs for all schemes to reduce training time and compute.
> 3. The difference between ResNet B and C on Tiny Imagenet for Kaiming initializations is significant but not that relevant, as Risotto achieves overall better results for both B and C. If Reviewer GPrm does not trust our findings, they are welcome to start their own runs to convince themselves of the correctness of our reports. We have shared our code and hyperparameter settings.
> 4. The reported learning rates were the result of careful hyperparameter tuning, as described in the appendix of our new draft. For instance, we observed that with a LR of 0.1 we achieve 56.94% accuracy on Tiny Imagenet but with a 0.01 LR we can reach 59.47 as reported in Table 2 for Risotto with BN. Hence, we choose 0.01 as the learning rate for Risotto.
> 5. As suggested by the reviewer, we have added experiments with ResNet B for the comparison with NF ResNets ( Figure 12(a)), BN in the last layer (Figure 12(b)) and training a deep 100 layer network (Figure 8(a)) in the Appendix of our updated manuscript. Additionally, we have also added experiments for a deeper 300 layer network (type B) to highlight the even stronger benefits of our method for deeper networks (see Figure 4(center)). Figure 13 in Appendix A.11 demonstrates that also a 2000 layer ResNet is trainable with Risotto on CIFAR100.
> The difference in the final accuracy between Fixup and Risotto on ResNet101 (See Figure 8(b) in Appendix A.6) is not significant. However, Risotto achieves earlier excellent results and trains faster. Note that Risotto wins also with respect to accuracy in a deeper ResNet model with 300 layers as updated in Figure 4(center).
> 6. We have added ImageNet experiments (see Figure 10 in Appendix A.8). After 120 epochs, a single run of ResNet 50 (B) with BN achieves 65.66% test accuracy for Risotto and 58.41 test accuracy for a baseline He initialization with BN. Without BN, ResNet 50 (B) achieves 67.05% test accuracy with Risotto, while Fixup achieves 67.5% test accuracy.
> Note that these findings are generally in line with our experiments on Tiny ImageNet. Usually, new initialization schemes are tested on large scale data and architectures because they only lead to effective improvements in this setting. Special about Risotto is that we can achieve improvements already in a smaller scale setting. We should not forget that many real world applications, in particular in the biomedical domain, are of this smaller scale. It is therefore highly relevant to improve deep learning algorithms in this setting as well. Note also that most researchers do not have the computational resources to train large scale models on huge datasets.

---

### Official Review · Reviewer_XaUD · 2022-10-27

**Confidence:** 4
**Correctness:** 3
**Technical Novelty And Significance:** 2
**Empirical Novelty And Significance:** 2
**Recommendation:** 3

**Clarity, Quality, Novelty And Reproducibility:**

Clarity:
The paper is mostly clear. I did not check any theorems or proofs

Quality:
The quality of the experimental section is low. The introduction/discussion is mostly clear but has some confusing/misleading statements.

Novelty:
The work is somewhat novel. The authors extend Looks Linear initialization from feedforward networks to ResNets, however they do not provide any discussion/experiments to motivate why this is necessary given that deep feedforward networks are already trainable.

Reproducibility:
The authors do not give full details describing their experiments (eg hyper-parameters, exact architecture).

Other comments:
1) The authors assert that feature diversity at initialization is important, however they don't provide any evidence for this. Additionally, do schemes satisfying dynamical isometry not by definition have very low feature diversity?

2) As i commented above, it is standard practice to not introduce an $\alpha$ or $\beta$ parameter, and consequently the scale of hidden activations on the forward path blows up exponentially without normalization and linearly with normalization. This is the basis of the argument in De and Smith as to how BN suppresses the residual branch.

3) does insight 2.6 assume that the network width is finite?

4) Note that NF-ResNets only require gradient clipping when training with large batch sizes, at small batch sizes gradient clipping isn't necessary as shown in Brock et al. (2021, a).

**Strength And Weaknesses:**

Strengths:

1) To my knowledge, this is the first paper to apply looks linear initialization (with orthogonal weights) to a ResNet.
2) Experiments suggest this scheme achieves good performance.

Weaknesses:

1) Note that Balduzzi et al. already combined orthogonal initialization with looks linear initialization in feedforward networks. Since this scheme already achieves dynamical isometry, it is not clear to me why the skip connection is necessary. However the paper does not explore this question? I would like to see experiments assessing whether or not the skip connection is beneficial in practice.

2) Definition 2.2 ("Normal ResNet initialization") is not how practitioners normally initialize ResNets. In particular, practitioners do not normally ensure $\alpha^2 + \beta^2 = 1$, indeed almost all implementations do not include either $\alpha$ or $\beta$. This leads to some confusing statements later in the text.

3) The experimental section is very weak. No large scale datasets (eg ImageNet) are included, and almost all experiments use the shallow ResNet-18 architecture, which is relatively easy to train with/without BatchNorm simply by ensuring the signal doesn't explode on the forward pass. Initialization becomes more important in deeper networks (eg 100+ layers).

4) No comment is given describing how learning rates and other hyper-parameters are tuned, and I suspect that poor tuning accounts for many of the observed differences between initialization schemes.

**Summary Of The Paper:**

The authors propose to initialize ResNets with ReLU activations with exact dynamical isometry by combining ideas from delta-orthogonal initialization and looks-linear initialization. The proposed initialization scheme guarantees that every residual block is an exact orthogonal map. A small empirical evaluation shows that the scheme is able to train moderately deep ResNets with/without batch normalization.

**Summary Of The Review:**

The proposed initialization scheme is quite complicated and it will only be adopted by community if there are clear empirical benefits. Since the experiments provided were not convincing, I recommend rejection.

---

> ### Author Response · Authors · 2022-11-13
> **Reply to Reviewer XaUD**
>
> We thank Reviewer XaUD for their feedback and critical analysis of our work.
> 1. Skip connections are popular for multiple reasons.
> Skip connections were not only introduced to increase the trainability of deep feed-forward architectures. They also change the way a neural network represents a function. Already in the original ResNet paper, He et al argued that residual layers might be easier to learn because they only need to represent potentially small nonlinear deviations from the identity function. Thus, ResNets could allow for simpler and sparser representations. This hypothesis is supported by the fact that only a single neuron per layer (but arbitrary depth) is required for the universal function approximation property of ResNets, as shown by [Lin and Jegelka](https://proceedings.neurips.cc/paper/2018/file/03bfc1d4783966c69cc6aef8247e0103-Paper.pdf). Furthermore, residual architectures seem to contain effective lottery tickets in contrast to VGGs without skip connections according to [Ma et al](https://openreview.net/pdf?id=WL7pr00_fnJ).
>
> 2. Our analysis covers practically relevant cases
> We have introduced $\alpha$ and $\beta$ to ease the comparison with other theoretical work and to discuss multiple approaches jointly. Initialization schemes that do not fulfill $\alpha^2 + \beta^2=1$ usually lead to vanishing and exploding gradients without normalization. Yet, normalization can also lead to effective choices of $\alpha$ and $\beta$ that make standard initialization set-ups fulfill $\alpha^2 + \beta^2=1$. For instance, in Kaiming normal initialization, which is commonly used with BN, the scaling of the residual and skip branches is effectively $\alpha = \beta = \sqrt{0.5}$. Fixup and SkipInit also fulfill our condition with $\alpha = 0$ and $\beta = 1$. We therefore cover the standard cases.
> 3. Experiments:
> We have added ImageNet experiments (see Figure 10 in Appendix A.8). After 120 epochs, a single run of ResNet 50 (B) achieves 65.66 test accuracy for Risotto and 58.41 test accuracy for a baseline He initialization. Without BN, ResNet 50 (B) achieves 67.05% test accuracy, while Fixup achieves 67.5% test accuracy. Note that the trend and hyperparameters are generally in line with our experiments on Tiny ImageNet.
> Usually, new initialization schemes are tested on large scale data and architectures because they only lead to effective improvements in this setting. Special about Risotto is that we can achieve improvements already in a smaller scale setting. Note that Risotto provides the on average best performing results on almost all datasets. The only exception is ResNet18 on CIFAR10 with BN. The likely reason is that ResNet18 is not very deep. On the other datasets, we usually train ResNet50 (and not ResNet18 as Reviewer XaUD claimed).
> To emphasize this point, we have complemented our experiments on ResNet101 with much deeper ResNets consisting of 300 layers (see Figure 4(center)). Our initialization Risotto trains faster than Fixup and achieves a higher test accuracy on CIFAR100. We also show that a 2000 layer Residual Network is trainable with Risotto on CIFAR100 (see Figure 13, in Appendix A.11).
> 4. Hyperparameter Tuning
> We followed a pretty standard hyperparameter tuning procedure separately for each initialization scheme + model + dataset. We split the training data randomly into 90% for training and 10% for validation to pick the hyperparameters based on the average of 2 independent runs (2 in case of Tiny ImageNet). For every experiment we use the best learning rate out of {0.1, 0.01, 0.05, 0.001} and decide between cosine annealing and step scheduling, while keeping the batch size fixed to 256. For the best performing parameters, we retrain on the full training data set and report results on the test data.
> Note that also our baselines achieve good results, which provides evidence for the fact that we tuned the hyperparameters fairly.

---

> > ### Author Response · Authors · 2022-11-13
> > **Reply to Reviewer XaUD (continued)**
> >
> > **Reproducibility**
> > We have provided all our code and the experimental setup of our experiments (see Appendix A.4). As mentioned, we also tuned the learning rate rigorously and reported the best results for each initialization scheme.
> >
> > **Feature Diversity**
> > As discussed, [Blumenfeld et al.](https://arxiv.org/abs/2007.01038) have analyzed the importance of feature diversity in detail and demonstrated that most ResNet initialization schemes (including Fixup) can be unstable for their lack of it. They test the hypothesis that the random ‘symmetry breaking` of diverse features leads to better training results. For instance, also in our case, Figure 3 (right) demonstrates that Risotto enables faster training.
> > As mentioned in Definition 2.2, Fixup and SkipInit set the residual branches to zero, limiting the contribution of the residual branch to feature diversity. In contrast, Risotto (ours) allows signal propagation through both branches. Figure 1 (c) highlights the benefits of our approach. Note that SkipInit corresponds to the choice $\alpha = 0$ (red).
> >
> > **Insight 2.6**
> > The insight holds in general for finite networks. However, as explicitly stated, to obtain the presented simplified relationship, we have used the mean field limit $N_m \rightarrow \infty$.
> >
> > **NFNETs** When we used sufficiently small batch sizes, we did not train with gradient clipping.
> >
> > **Usability**
> > Our initialization scheme consists only of a few lines of code. Reviewer VqzH found it simple and easy to reproduce.
> > It can be easily adapted to existing networks in practice. Furthermore, it would require only one line of code to import our initialization scheme into an existing model from the code we have provided.

---

> > > ### Comment · Reviewer_XaUD · 2022-11-22
> > > **clarifications/further questions**
> > >
> > > I thank the reviewers for their response.
> > >
> > > First a clarification; I had not realized in my initial review that the authors primarily study post-activation ResNets (ResNet-V1), I had assumed that the ResNet-B networks were pre-activation/ResNet-V2. My apologies for this.
> > >
> > > This explains the discrepancy regarding the claim in De and Smith that BN suppresses the residual branch, since De and Smith only study pre-activation ResNets, on which the scale of the signal at the end of each residual block grows linearly with depth. This may explain why SkipInit and NF-ResNets perform poorly in some experiments, since both schemes were designed for pre-activation ResNets. Fixup is the only baseline in the paper which was designed for post-activation ResNets (although it also performs better on pre-activation ResNets in practice).
> > >
> > > Further questions/comments:
> > >
> > > 1) Note that although post-activation ResNets are often used as baselines, to my knowledge essentially all ongoing research (especially at scale) uses pre-activation ResNets, specifically because they perform much better as depth increases (since it is far easier to achieve initializations close to dynamical isometry).
> > >
> > > 2) Why is it desirable to have balanced skip/residual connections? As the authors state above, the benefit of ResNets is that it allows the residual branch to learn small transformations. If the skip and residual connections are balanced, do we not effectively have a feedforward networks with slightly reduced effective depth? This is why I think it is necessary to also compare to feedforward networks in the experimental section, to confirm that there is a benefit to the presence of skip connections after imposing dynamical isometry.
> > >
> > > 3) In the experiments, the authors state that alpha is initialized to 1. If alpha^2 + beta^2 = 1, does this mean that the skip connection is ignored at initialization?
> > >
> > > 4) I'd encourage the authors to clarify the distinction between pre and post activation ResNets in the text, and make clear which methods were designed for which networks.
> > >
> > > 5) I am still concerned by the experimental section. It does not present convincing evidence that the scheme outperforms previous methods.

---

> > > > ### Author Response · Authors · 2022-11-26
> > > > **Clarification about Pre-Activation ResNets and additional experiments**
> > > >
> > > > We thank the reviewer for pointing out the different implementations of ResNets used by SkipInit, NF ResNets and Fixup.
> > > > In light of this, we have already added experiments to show that Risotto is well trainable on Pre- and Post-Activation ResNets.
> > > >
> > > > 1. The reason why we use Post-Activation ResNets is because our main baseline, Fixup, was conducted in this setting and we wanted to be consistent in all our experiments. Still, we have demonstrated in additional experiments that Pre-Activation ResNets also benefit from Risotto initialization as shown in Appendix A.10. In addition, we have repeated the experiments for Figure 2c) for different values of $\alpha$ for Pre-Activation networks (see below).
> > > >
> > > > 2. Figure 2c) points out empirically that balanced activations seem to have a benefit for the performance of ResNets in combination with BN. We have repeated these experiments for Post-Activation ResNets and found the same trends. It is correct that convolutional feedforward architectures without skip connections with orthogonal looks-linear initialization represent a similar function initially. However, as discussed in our last response, the function representation, training dynamics, etc. are still quite different for ResNets. It is therefore not surprising that residual architectures will sometimes achieve better results than networks without skip connections. To give an example, we have trained a ResNet50 without any skip connections (just a feedforward network) on CIFAR100 and achieved an accuracy of 52% across three runs while the equivalent accuracy of the same ResNet with skip connection is 78% as shown in Table 2. This result confirms that skip connections can have benefits even in case of initial dynamical isometry.
> > > >
> > > > 3. Unfortunately, we have overloaded the notation $\alpha$ in Definition 2.2 versus 2.8 and 2.9, which we will correct. In Definition 2.8 and 2.9, we use alpha = 1 to denote the case that the skip and the residual branch receive identical weight and are thus balanced. While Figure 2(c) shows that alpha >= 1 (meaning balanced contributions of residual and skip connections) leads to high performance, we also point out in Definition 2.8 that Risotto can enable dynamical isometry for any value of alpha. Our experiments without BN confirm that Pre- as well as Post-Activation ResNets C and B are all well trainable for a wide range of values of $\alpha$.
> > > >
> > > > 4. We would be happy to clarify the type of ResNet that was used in our experiments and add our additional results on Post-Activation ResNets to the appendix.
> > > > We would also like to emphasize that while each of the methods we compare to (like Fixup and SkipInit) were designed for one specific ResNet type, our method is flexible regarding the type and induces dynamical isometry in both cases.
> > > >
> > > > 5. As we have shown through extensive experiments on multiple datasets, Risotto is flexible and is able to match or outperform He initialization (with BN) and Fixup and SkipInit (without BN). Risotto clearly outperforms other methods on Tiny ImageNet. In addition, it enables training very deep networks like a 2000 layer network.
> > > > Furthermore, we have added experiments on ImageNet, which show that Risotto achieves more robust performance across a range of training schedules. In response to Reviewer VqzH, we have further shown that Risotto is able to achieve an  accuracy of 75.7% on ImageNet (using a ResNet50 (B) with BN) when trained in an experimental setup that was tuned for our baselines. These results confirm that Risotto can indeed be used as a drop-in replacement for existing initialization schemes without the need for extensive hyperparameter tuning.

---

### Official Review · Reviewer_MPAw · 2022-11-02

**Confidence:** 2
**Correctness:** 3
**Technical Novelty And Significance:** 3
**Empirical Novelty And Significance:** 3
**Recommendation:** 6

**Clarity, Quality, Novelty And Reproducibility:**

The paper is well written and easy to follow. For related works, I suggest the authors to disentangle the proposed method from others such as Fixup and SkipInit more. Currently it is not clear why is the main contribution of your method that differentiates it from the previous works.

**Strength And Weaknesses:**

Strengths:
1. The derived signal propagation results does not require mean field approximation but highlight input separability issues.
2. The proposed method considers the contribution from both residual and skip branches unlike most previous works achieving dynamical isometry in residual networks.
3. Promising empirical results on CIFAIR and Tiny ImageNet datasets.

Weaknesses:
1. It is not clear why feature diversity is important for the initialization. The authors differ the proposed method from the previous works like Fixup or SkipInit by arguing that the proposed method promotes feature diversity. However, the reason why feature diversity is needed is not well motivated in the paper. It would be good to study how the feature diversity affects the initialization.
2. The proposed method has limited applicability. Because the method and its derivation highly rely on post-activation ResNet structure, it can not be directly applied to pre-activation ResNet (which could achieve dynamical isometry easily using skip connections). As it is limited to residual networks, it can not be applied to other models such as transformer.
3. Large-scale evaluations are needed to show the effectiveness of the proposed method. The authors only evaluate the proposed method on CIFAIR and Tiny ImageNet datasets. It would be good to evaluate the method on full ImageNet setting.

**Summary Of The Paper:**

The authors propose a new random initialization scheme Risotto that achieves dynamical isometry for residual networks with Relu activation functions. The key difference from the previous work is that they achieve the dynamical isometry by balancing the signals from both residual and skip branches, unlike the previous works that mainly suppress the signal from the residual branch. They provide theoretical justification for the proposed method and extensive empirical experiments to denote the effectiveness of the proposed method.

**Summary Of The Review:**

Overall I believe the paper is above the acceptance level, and I would like to raise my score if the authors address the above concerns.

---

> ### Author Response · Authors · 2022-11-13
> **Reply to Reviewer MPAw**
>
> We thank Reviewer MPAw for their in-depth review and are happy to address their concerns as follows.
>
> 1. Feature diversity
> [Blumenfeld et al.](https://arxiv.org/abs/2007.01038) have analyzed the importance of feature diversity in detail and demonstrated that most ResNet initialization schemes (including Fixup) can be unstable for their lack of it. They test the hypothesis that the random ‘symmetry breaking` of diverse features leads to better training results. For instance, Figure 4 (center) demonstrates that Risotto enables faster training.
> Furthermore, both Fixup and SkipInit scale down residual connections relative to the skip connections. In contrast, our initialization scheme Risotto promotes signal propagation through both parts initially. Figure 2 (c) highlights the benefits of our approach. Note that SkipInit corresponds to the choice $\alpha = 0$ (red).
>
> 2. Risotto is widely applicable and works for pre-activation and post-activation ResNets
> Our initialization scheme Risotto also applies to pre-activation ResNets, as the looks-linear structure is flexible with regard to the position of the ReLU. We have added an experiment with a pre-activation ResNet to demonstrate the broad applicability of our proposal (see Figure 11 in Appendix A.9). The only reason why we have focused on post-activation blocks is that they seem to be more popular, as they are available in PyTorch default models.
> Risotto could also improve the initialization of transformers by following a similar approach as ReZero by Bachlechner et al. and scaling down additional transformer edges. However, this is out of scope of our work, since our whole theory and exposition is focused on ResNets, which are still quite popular deep learning architectures.
>
> 3. Large Scale Experiments
> We have added ImageNet experiments (see Figure 10 in Appendix A.8). After 120 epochs, a single run of ResNet 50 (B) achieves 65.66% test accuracy for Risotto and 58.41 test accuracy for a baseline He initialization with BN.
> Without BN, ResNet 50 (B) achieves 67.05% test accuracy, while Fixup achieves 67.5% test accuracy after 100 epochs.
> Note that these findings are generally in line with our experiments on Tiny ImageNet. Usually, new initialization schemes are tested on large scale data and architectures because they only lead to effective improvements in this setting. Special about Risotto is that we can achieve improvements already in a smaller scale setting.
> In response to Reviewer XaUD, we have complemented our experiments on much deeper ResNets consisting of 300 layers (see Figure 4(center) and also Appendix A.6). Our initialization Risotto trains faster than Fixup and achieves also a higher test accuracy on CIFAR100 (see Figure 4(center)). We also show that a 2000 layer Residual Network is trainable with Risotto on CIFAR100 (see Figure 13, in Appendix A.11).

---

### Author Response · Authors · 2022-11-13
**Details of revised manuscript**

We thank the reviewers for their detailed feedback. Based on their feedback we have done the following changes and additional experiments:

1. Introduction
- We have added figures (Figure 1 and 3) and adapted the introduction to explain the main features of our initialization scheme Risotto i.e. feature diversity, balancedness and dynamical isometry.

2. Experiments
- ImageNet: We report the following results on ImageNet for a single run as permitted with our limited compute. For ResNet50 (B) with BN, Risotto achieves 65.66% test accuracy and a baseline He initialization achieves 58.41 test accuracy. Without BN, ResNet 50 (B) achieves 67.05% test accuracy with Risotto, while Fixup achieves 67.5% test accuracy.
- We show in Figure 13 Appendix A.11 that a 2000 layer residual network is trainable with Risotto.
- In Figure 4 (center) we show that Risotto trains faster than Fixup in a 300 layer ResNet without BN.
- We also provide additional experiments with ResNet (B) architectures to verify the usability of Risotto in the Appendix (A.6-A.11).
- We show that Risotto can also be transferred to Pre-Activation ResNets (Figure 11 in Appendix A.9)

---

### Author Response · Authors · 2022-11-18
**General Response**

We thank all the reviewers for their constructive suggestions and detailed feedback.

In response, we have added two figures to clarify our contributions and conducted the following additional experiments:
1. We have trained ResNet50 models with Risotto on ImageNet with and without BN. Our results compare favorably with the respective baselines.
2. We have demonstrated that Risotto enables training very deep neural networks by initializing and training a 2000 layer residual network on CIFAR100.
3. We have verified in experiments that Risotto also applies to preactivation ResNets.
4. We also provided additional experiments with ResNet (B) about the BN layer placement, comparison with NF ResNets and training a 300 layer network without BN.

We believe that we have addressed all points of criticism but would welcome discussions. We would be happy to provide further clarifications and information in case of any misunderstandings.

In summary, we propose Risotto, an initialization scheme for deep residual neural networks that achieves perfect dynamical isometry, feature diversity, and balances skip as well as residual connections. We provide theoretical as well as experimental evidence that Risotto often improves the trainability of residual layers with and without batch normalization.
As residual layers are popular building blocks of state-of-the-art neural network architectures, we are confident that we provide a valuable contribution that is of interest to the ICLR community.

---

### Author Response · Authors · 2022-12-08
**Gentle Reminder**

Since the discussion period is now coming to an end, we would like to thank the reviewers for their feedback and request them to follow up on our reply to the same.

We believe that we have addressed all points of criticism and have accordingly included additional experiments and revised our manuscript as per the suggestions of the reviewers.
We would be happy to provide further clarifications and engage in further discussion.

To summarize, our initialization scheme Risotto, is able to achieve exact dynamical isometry for residual networks and enable successful training on a wide range of datasets including but not limited to ImageNet.
This simple initialization scheme can be a useful addition to the toolkit of any deep learning practitioner using residual networks and would be a valuable contribution to the ICLR community.

We would be grateful if the reviewers would acknowledge our comments and increase their scores accordingly.

---

### Decision · Program_Chairs · 2023-01-20

**Decision:**

Reject

**Justification For Why Not Higher Score:**

While the paper does develop an initialization scheme for residual networks with strong theoretical motivation and precedence for success, there is ultimately not enough evidence of its practical utility to merit publication at this time.

**Justification For Why Not Lower Score:**

N/A.

**Metareview: Summary, Strengths And Weaknesses:**

This paper introduces a novel initialization scheme (Risotto) for residual networks with ReLU activations that achieves exact dynamical isometry, namely the input-output Jacobian has all its singular values equal to 1. Experiments show modest improvements over baselines in some settings.

The main contribution of this paper is Theorem 2.10/2.11, which shows that Risotto achieves exact dynamical isometry for certain residual network architectures. Risotto is an important contribution to the line of work that had previously found promising benefits to dynamical isometry in simpler architectures. In my view, the reviews do not adequately acknowledge this important contribution.

However, the reviewers do highlight a number of weaknesses with the paper. The motivation and framing in the context of prior work need improvement, and the experimental evaluation is not convincing. Because the main ideas behind the initialization scheme are not novel, and most of the constituent pieces have been analyzed elsewhere, the novelty of this work mainly rests on the specifics of the Risotto initialization and, ultimately, how well it works. The experimental results show some promise, but the evaluation is not sufficiently thorough and the practical impact of the method remains in question.